# FEDERATED AGENT REINFORCEMENT LEARNING

## ABSTRACT

Autonomous AI Agents powered by LLMs have shown remarkable abilities in diverse domains. However, the training process typically require *centralized* collection of large amounts of real-world user data, posing substantial privacy and regulatory concerns. To this end, we explore a new *decentralized* training paradigm, namely **FEDAGENT** (Federated Agent Reinforcement Learning), which enables collaborative learning of AI agents across distributed clients without sharing local data. Moreover, we construct the first decentralized agent learning environment **FEDAGENTGYM**, which includes four types of LLM agents, two application scenarios (WebShop and ALFWorld), three variations of decentralized settings, and three newly defined heterogeneity challenges (*Preference Heterogeneity*, *Coverage Heterogeneity*, and *Hardness Heterogeneity*), to systematically investigate its effectiveness and impact factors. Extensive theoretical and empirical studies show that FEDAGENT can have comparable performance to the centralized training paradigm and exhibit strong robustness against heterogeneities, which shows the feasibility of training AI agents without sacrificing data privacy. The code is available here.

## 1 INTRODUCTION

The rapid advancement of AI agents, especially those powered by Large Language Models (LLMs), has demonstrated remarkable capabilities across diverse domains, from web navigation to embodied environments (Zhang et al., 2025; Gao et al., 2025; Liu et al., 2025). However, training these agents typically requires *centralized* access to vast amounts of users' real-world task query and trajectory data, which are inherently privacy-sensitive and hard to acquire due to regulatory compliance. Thus, a foundational question is: ***how to train AI agents while protecting users' data privacy?***

In this paper, we explore a new *decentralized* training paradigm, namely **FEDAGENT** (**Federated Agent Reinforcement Learning**), which enables collaborative learning of AI agents, particularly LLMs, across distributed clients without sharing local data. In each round, the server distributes the current model to selected clients, who then train locally on their own data and send back their updated models. The server aggregates these updates by averaging them to create an improved global model for the next round. This process repeats iteratively, facilitating distributed LLM agent training while preserving data privacy since only model parameters, not raw data, are exchanged.

Compared with the previous federated learning literature, FEDAGENT is faced with fundamentally new challenges. The majority of existing federated learning research has concentrated on supervised classification tasks. There are also recent works that have explored federated reinforcement learning (FRL) for traditional RL settings (Liu et al., 2024; Qi et al., 2021; Kairouz et al., 2021). However, both of them operate under distinct assumptions compared to LLM agent learning. Supervised federated learning is usually built on static data distributions and one-shot predictions, while traditional FRL typically assumes simple rewards, well-defined state and action spaces. In contrast, LLM agent learning involves *diverse task formulations, multi-step natural language reasoning, and complex environment interactions*, which create entirely new challenges for federated paradigms.

To systematically investigate the effectiveness of this new training paradigm as well as the impact factors, we built the first decentralized agent learning environment **FEDAGENTGYM**, which incorporate four types of LLM agents (Qwen2.5-{1.5,3,7}B-Instruct and Llama-3.2-3B-Instruct), two applications (WebShop and ALFWorld), three variations of decentralized settings (samples per client, clients selected per communication round, and local training epochs per client per round).

Importantly, since the existing heterogeneity challenges in federated learning have mostly been defined in the context of supervised classification tasks (Ye et al., 2023; Gao et al., 2022), which

Figure 1: An Illustration of FEDAGENT and FEDAGENTGYM.

focus on label skew, feature shift, or quantity imbalance, they are not directly applicable to LLM agent learning. Thus, we propose three new and orthogonal definitions of client heterogeneity unique to decentralized AI agent learning: ***Preference Heterogeneity***, where clients may prefer distinct types of tasks; ***Coverage Heterogeneity***, where the task sampling scope may vary across clients; ***Hardness Heterogeneity***, where the overall difficulty of tasks may differ among clients. Moreover, we carefully design three novel client partitioning strategies PREFERENCEPARTITION, COVERAGEPARTITION, and HARDNESSPARTITION accordingly, grounded in mathematical techniques such as *Gaussian Noise*, *Multinomial Sampling* and *Beta Distribution*. These strategies allow us to precisely control the extent of one type of heterogeneity across clients with a single hyperparameter, while keeping the other characteristics of the client distribution unchanged. We then incorporate them into FEDAGENTGYM to isolate and analyze the impact of each form of heterogeneity on FEDAGENT separately.

To validate the effectiveness of FEDAGENT, we first conduct a theoretical analysis on its convergence. Then, through extensive and systematic empirical studies with FEDAGENTGYM, we demonstrate that FEDAGENT consistently outperforms *local agent training* and can achieve performance comparable with *centralized agent training*, despite never sharing local data. Moreover, FEDAGENT exhibits strong robustness to the aforementioned preference, coverage, and hardness heterogeneity challenges, while revealing sensitivities to certain decentralized configurations. Overall, our studies show the potential of scalable agent learning without sacrificing data privacy, provide valuable insights that inform practical deployment, and open new research directions in the field of agent learning.

Our contributions can be summarized as follows:

- We explored a new decentralized paradigm of training AI agents, namely **FEDAGENT** (**Federated Agent Reinforcement Learning**), which enables collaborative agent learning across distributed clients without sharing local data. We also provide a theoretical analysis on its convergence.

- We propose to categorize the new client heterogeneity challenges in decentralized agent learning into *Preference Heterogeneity*, *Coverage Heterogeneity*, and *Hardness Heterogeneity*. To investigate how each type of heterogeneity affects the performance, we introduce three novel client partitioning methods: PREFERENCEPARTITION, COVERAGEPARTITION, and HARDNESSPARTITION.

- We constructed the first decentralized agent learning environment **FEDAGENTGYM**, which includes four types of LLM agents, two applications (WebShop and ALFWorld), three variations of decentralized settings, and three heterogeneity challenges, to analyze the performance of FEDAGENT systematically and controllably, and offer insights to guide future development.

- Extensive studies show that FEDAGENT not only beats the single-client local training paradigm but also can achieve comparable performance to the centralized agent learning paradigm. Furthermore, FEDAGENT shows high robustness against the three types of heterogeneity challenges. We also provide insights on its sensitivity to different decentralized settings.

- We release our code and environment as an extendable open-source library to inspire more future works in this new direction. The link to the repository is available here.

---

**Algorithm 1** FEDAGENT with Client and Server training

---

**Require:** Total clients $K$, rounds $T$, clients-per-round $M$, local steps $\tau$, learning rate $\eta$
**Ensure:** Final LLM-based global policy parameters $\theta_{\text{final}}$
 1: Initialize global policy parameters $\theta_0$ (an LLM)
 2: **for** $t = 0$ **to** $T - 1$ **do**
 3:     **Server:** sample client subset $S_t \subset [K]$ with $|S_t| = M$ (uniform without replacement)
 4:     **Server:** broadcast $\theta_t$ to all $k \in S_t$
 5:     **for each** $k \in S_t$ **in parallel do**
 6:         Set local iterate $\theta_{k,t,0} \leftarrow \theta_t$
 7:         **for** $i = 0$ **to** $\tau - 1$ **do**
 8:             Collect a mini batch of trajectories $B_{k,t,i}$ using policy $\pi_{\theta_{k,t,i}}$ in environment $\mathcal{M}_k$
 9:             Estimate policy gradient for $J_k(\theta_{k,t,i})$ on client $k$:

$$g_{k,t,i} \leftarrow \nabla_\theta \hat{J}_k(\theta_{k,t,i}; B_{k,t,i}) \quad (e.g., \text{GRPO})$$

 10:             Local update: $\theta_{k,t,i+1} \leftarrow \theta_{k,t,i} + \eta\, g_{k,t,i}$
 11:         **end for**
 12:         Client returns local model $\theta_{k,t,\tau}$                    ▷ equivalently $\Delta\theta_{k,t} = \theta_{k,t,\tau} - \theta_t$
 13:     **end for**
 14:     **Server:** Aggregation via model averaging:

$$\theta_{t+1} \leftarrow \frac{1}{M} \sum_{k \in S_t} \theta_{k,t,\tau} \quad (\text{equivalently } \theta_{t+1} = \theta_t + \tfrac{1}{M} \sum_{k \in S_t} (\theta_{k,t,\tau} - \theta_t))$$

15: **end for**
16: **return** $\theta_{\text{final}} \leftarrow \theta_T$

---

## 2 FEDAGENT: FEDERATED AGENT REINFORCEMENT LEARNING

As shown in Algorithm 1, we consider a federated reinforcement learning setup for FEDAGENT. A population of clients are indexed by $k \in [K] = \{0, \ldots, K-1\}$. Training proceeds for communication rounds $t = 0, \ldots, T - 1$. At round $t$, the server samples a subset $S_t \subset [K]$ of size $|S_t| = M$ uniformly without replacement, broadcasts the current global policy parameters $\theta_t$, and aggregates the participating clients' locally updated parameters.

**LLM Agent Training.** The agent is a parametric policy $\pi_\theta$ (an LLM) that, conditioned on a task description $c$ and an interaction history $h_u$ up to step $u$, produces an action $a_u \sim \pi_\theta(\cdot \mid h_u, c)$. An action often contains both *a sequence of free-form tokens* (*i.e.*, the agent's intermediate reasoning) and *environment-facing choice* (*e.g.*, tool API calling). Each client $k$ operates in a Markov Decision Process (MDP) environment $\mathcal{M}_k = (\mathcal{S}_k, \mathcal{A}_k, P_k, r_k, \rho_k, \gamma)$ with state space $\mathcal{S}_k$, action space $\mathcal{A}_k$, transition kernel $P_k$, reward function $r_k$, initial-state distribution $\rho_k$, and discount $\gamma \in (0, 1]$. Client $k$ also has a distribution $\mathcal{D}_k$ over textual task descriptions $c \in \mathcal{C}_k$. Fix $k$ and a task description $c \sim \mathcal{D}_k$. The agent interacts with $\mathcal{M}_k$ for horizon $H$, producing a trajectory:

$$\chi = (c, s_0, a_0, r_0, \ldots, s_H), \quad s_0 \sim \rho_k(\cdot \mid c), \ a_u \sim \pi_\theta(\cdot \mid h_u, c), \ s_{u+1} \sim P_k(\cdot \mid s_u, a_u, c).$$

The discounted return of $\chi$ is $R(\chi) = \sum_{u=0}^{H-1} \gamma^u r_u$. It is worth noting that when the LLM agents generate $H$ consecutive textual actions $(a_0, ..., a_{(H-1)})$ in a trajectory $\chi$, each action may span thousands of tokens, considering LLM agents' long reasoning capacity (DeepSeek-AI et al., 2025). This makes token-level credit assignment across the trajectory particularly challenging.

**Local objective (client $k$).** Client $k$ aims to maximize the expected episodic return of the policy on its own environments and tasks:

$$J_k(\theta) = \mathbb{E}_{c \sim \mathcal{D}_k} \mathbb{E}_{\chi \sim (\pi_\theta, \mathcal{M}_k, c)} [R(\chi)]. \tag{1}$$

During round $t$, each participating client initializes a local iterate at the broadcast model, $\theta_{k,t,0} \leftarrow \theta_t$, and performs $\tau$ steps of stochastic policy optimization. At local step $i \in \{0, \ldots, \tau - 1\}$, the client collects a batch of trajectories $B_{k,t,i} = \{\chi^{(b)}\}_{b=1}^{N_{k,t,i}}$ by interacting with $\mathcal{M}_k$ under $\pi_{\theta_{k,t,i}}$ and computes a policy-gradient estimate:

$$g_{k,t,i} = \nabla_\theta \widehat{J}_k(\theta_{k,t,i}; B_{k,t,i}) = \frac{1}{N_{k,t,i}} \sum_{b=1}^{N_{k,t,i}} \sum_{u=0}^{H(\chi^{(b)})-1} \nabla_\theta \log \pi_{\theta_{k,t,i}}\big(a_u^{(b)} \mid h_u^{(b)}, c^{(b)}\big) \widehat{A}_u^{(b)}, \quad (2)$$

where $\widehat{A}_u^{(b)}$ is any valid return/advantage signal (*e.g.*, a GRPO-style estimator (Shao et al., 2024)). The local update is $\theta_{k,t,i+1} = \theta_{k,t,i} + \eta\, g_{k,t,i}, (i = 0, \dots, \tau - 1)$ with step size $\eta > 0$. After $\tau$ steps the client returns its local model $\theta_{k,t,\tau}$ (equivalently the update $\Delta\theta_{k,t} = \theta_{k,t,\tau} - \theta_t$) to the server.

**Global objective and aggregation.** The federated learning goal is to maximize a weighted average of client objectives:

$$J_{\text{global}}(\theta) = \sum_{k=0}^{K-1} w_k\, J_k(\theta), \qquad w_k \geq 0, \quad \sum_{k=0}^{K-1} w_k = 1. \quad (3)$$

In the FEDAGENT, the server uses uniform model averaging over the $M$ participating clients each round (i.e., $w_k = \frac{1}{K}$ conceptually, with partial participation realized by $S_t$). Upon receiving the $\tau$-step local models $\{\theta_{k,t,\tau}\}_{k \in S_t}$, the server performs model averaging: $\theta_{t+1} = \frac{1}{M} \sum_{k \in S_t} \theta_{k,t,\tau} = \theta_t + \frac{1}{M} \sum_{k \in S_t} \big(\theta_{k,t,\tau} - \theta_t\big)$. After $T$ rounds the server outputs $\theta_{\text{final}} = \theta_T$.

## 3 FEDAGENTGYM: A DECENTRALIZED AGENT LEARNING ENVIRONMENT

### 3.1 LLM AGENTS AND APPLICATION DATASETS

FEDAGENTGYM is designed as an environment to investigate the impact factors of training AI agents, especially LLM, in a decentralized way. It includes four types of LLM agents, including `Qwen2.5-{1.5,3,7}B-Instruct` and `Llama-3.2-3B-Instruct`, and two challenging application datasets (WebShop (Yao et al., 2022) and ALFWorld (Shridhar et al., 2020)), which require complex reasoning process and multi-step environment interactions. We adopt these two datasets to simulate the real-world scenarios where data privacy concerns are paramount.

WebShop is a web-based interactive platform that evaluates LLM agents within authentic e-commerce scenarios. Task completion requires agents to navigate a simulated HTML shopping interface to locate, browse, and purchase appropriate items. The dataset features an extensive catalog of over $1.1$ million products paired with $12,000$ user instructions, creating a rich and varied action space.

ALFWorld provides an embodied simulation benchmark that evaluates LLM agents' capacity for sequential decision-making tasks. Each scenario presents the agent with a textual objective that must be achieved through iterative environment interaction. The dataset encompasses $3,827$ task instances spanning six types of household activities: Pick & Place (Pick), Examine in Light (Look), Clean & Place (Clean), Heat & Place (Heat), Cool & Place (Cool), and Pick Two & Place (Pick2).

### 3.2 DECENTRALIZED SETTINGS

We comprehensively examine the impact of different decentralized settings on FEDAGENT performance across three critical dimensions. First, we vary **the number of samples per client**, which determines the sampling scope for each LLM agent's exploration of the action space and response generation, directly affecting both the diversity of experiences collected and the quality of policy gradient estimates. Second, we change **the number of clients selected per communication round**, controlling both the computational parallelism and the degree of heterogeneity in exploration strategies incorporated during global model aggregation. Third, we adjust **the number of local training batches per client per round**, governing the extent of local optimization on the sampled trajectories before synchronization with the central server. These parameters collectively influence fundamental trade-offs between exploration diversity, communication overhead, and convergence stability in the federated setting. Through extensive studies across these dimensions, we characterize how different decentralized training design choices affect the final policy performance of FEDAGENT.

### 3.3 HETEROGENEITY CHALLENGES

To systematically evaluate how FEDAGENT performs under realistic client distributions, we propose three novel and orthogonal heterogeneity definitions, as conventional heterogeneity dimensions in federated classification tasks (*e.g.*, feature or label skew) (Ye et al., 2023; Gao et al., 2022) are not

directly applicable. We also propose the corresponding client partitioning strategies, allowing us to understand the individual impact of different heterogeneity types separately.

**Preference Heterogeneity: When Clients Have Different Task Preferences.** In real-world federated learning, different clients often *prefer distinct types of tasks*. For example, in the ALFWorld, some users might frequently interact with kitchen-related tasks (like "put the apple in the fridge"), while others primarily encounter bedroom tasks (like "examine the lamp"). In WebShop, some may have mostly electronics searches while others mainly focus on clothing or home goods.

To simulate this preference heterogeneity, we propose the PREFERENCEPARTITION algorithm. The pseudo code is illustrated in Algorithm 2 in Appendix B.1. We model this by starting with the global distribution of task categories and introducing controlled noise to create client-specific preferences. Specifically, we add **Gaussian Noise** to the log-probabilities of the global category distribution, apply softmax normalization, and use the resulting probabilities to sample $L$ instructions per client via **Multinomial Sampling**. This approach allows precise control over client distributions with a hyperparameter $\omega$ on topical preference heterogeneity, while maintaining the same total dataset size and per-client instruction count. More specifically, small noise values produce clients with similar task distributions, while larger noise creates highly specialized clients with distinct preferences.

**Coverage Heterogeneity: When Clients Have Different Task Sampling Scopes.** Even when clients encounter similar types of tasks, they may face vastly different quantities. A larger quantity of tasks indicates *coverage of a broader sampling scope per epoch* in reinforcement learning (we follow the setting in (Feng et al., 2025) to iteratively sample with replacement from the local data each epoch), while the sampling size remains fixed. Importantly, this differs from the quantity imbalance in conventional supervised federated classification tasks, where *training proceeds over the entire dataset each epoch*. In WebShop, for instance, some users might have extensive browsing histories with hundreds of product interactions, while others have only completed a few shopping sessions.

To model this coverage heterogeneity, we develop the COVERAGEPARTITION algorithm. The pseudo code is shown in Algorithm 3. We fix a global overlap target r (representing the average number of clients that see each instruction) and draw each client's data quantity from a **Beta Distribution**, which we then map to the range $[L_{\min}, L_{\max}]$. Task instructions are allocated to clients using weighted sampling without replacement to satisfy both individual client quotas and the global overlap constraint. This method isolates the effect of task sampling scope on FEDAGENT performance while keeping the underlying task distribution consistent across clients. Also, this method controls the extent of coverage heterogeneity via hyperparameter $\xi$ without impacting the overall mean of client quantities.

**Hardness Heterogeneity: When Clients Face Different Task Difficulties.** A particularly important but often overlooked source of heterogeneity is the overall difficulty of tasks that different clients encounter, which can be quantified by the *success rate* of tasks. For example, in ALFWorld, some clients might consistently face simple navigation tasks with high success rates, while others encounter complex multi-step reasoning tasks that frequently result in failure.

As demonstrated in Algorithm 4, our proposed HARDNESSPARTITION algorithm addresses this by partitioning the task instruction pool into "successful" and "unsuccessful" examples with a pretrained checkpoint. Then, using our COVERAGEPARTITION method, we first distribute successful instructions according to a **Beta Distribution** that determines each client's success rate. We then fill remaining slots with unsuccessful examples sampled uniformly, ensuring all clients have exactly $L$ instructions. This method enables us to study how different success rate distributions, which are controlled by a hyperparameter $\xi'$ and measures the extent of hardness of task distributions for each client, affect FEDAGENT while maintaining consistent dataset sizes and global overlap patterns across all clients.

## 4 THEORETICAL ANALYSIS ON CONVERGENCE

**Theorem 1** (Convergence of FEDAGENT). *Under Assumptions 1–5, suppose that at each communication round $t$ the server uniformly samples without replacement a subset $S_t \subset [K]$ of size $M \leq K$ and aggregates only those clients' updates:$\theta_{t+1} = \theta_t + \frac{1}{M} \sum_{k \in S_t} \Delta\theta_{k,t}$, with the same local inner loop and notation as in Algorithm 1. Let each selected client perform $\tau$ local steps with stepsize $\eta$. Choose the stepsize $\eta = \frac{1}{L\tau}$ and let $\tilde{\theta}$ be a uniform random iterate drawn from $\{\theta_t\}_{t=0}^{T-1}$. Then*

$$\mathbb{E}\left[J(\theta^\star) - J(\tilde{\theta})\right] \leq \frac{L}{\mu T}\left(J(\theta^\star) - J(\theta_0)\right) + \frac{1}{2\mu}\left[\frac{G^2 + \sigma^2}{M} + \frac{2(K-M)}{M(K-1)}\zeta^2 + \frac{(\tau-1)^2}{2\tau^2}\left(G^2 + \sigma^2\right)\right].$$

*In particular, the convergence rate remains $O(1/T)$. The "noise floor" comprises: (i) an $O(1/M)$ local stochastic term, (ii) an $O\left(\frac{K-M}{M(K-1)}\right)$ heterogeneity term due to client sampling, and (iii) an $O\left(\frac{(\tau-1)^2}{\tau^2}\right)$ local-drift term that vanishes when $\tau = 1$.*

**Remark.** Theorem 1 establishes that the FEDAGENT paradigm converges to a neighborhood of the optimum under the PL and smoothness conditions. With stepsize $\eta = 1/(L\tau)$ and $T$ communication rounds, the suboptimality decomposes into (i) a transient term that decays as $O(1/T)$ and (ii) a time-invariant "noise floor". The floor tightens with a larger $M$, vanishes in the full-participation and single-step homogeneous limit ($M = K$, $\tau = 1$, $\zeta^2 \approx 0$), and otherwise quantifies the computation–communication trade-off. The proof of Theorem 1 is in Appendix C. The key implications are:

1. **Convergence rate ($O(1/T)$):** The term $\frac{L}{\mu T}(J(\theta^*) - J(\theta_0))$ exhibits a linear-in-$1/T$ convergence rate with respect to the number of communication rounds. Better conditioning (smaller $L/\mu$) accelerates approach towards the asymptotic regime.

2. **Effect of partial participation ($O(1/M)$):** The variance term $\frac{G^2+\sigma^2}{M}$ decays inversely with the number of participating clients each round. Increasing $M$ reduces stochastic noise in the aggregated update. In the limit $M = K$, it matches the variance level under full participation.

3. **Client sampling and heterogeneity ($O\left(\frac{K-M}{M(K-1)}\right)$):** The middle term $\frac{K-M}{M(K-1)}\zeta^2$ is induced by client sampling each round without replacement under the heterogeneity assumption. It vanishes when $M = K$ and grows with smaller $M$ and larger heterogeneity $\zeta^2$, implying the potential benefits of a larger number of clients each round, stratified or clustered client sampling.

4. **Local-drift from multiple local steps ($O\left(\frac{(\tau-1)^2}{\tau^2}\right)$):** Performing $\tau > 1$ local steps introduces a bias captured by $\frac{(\tau-1)^2}{2\tau^2}(G^2 + \sigma^2)$. This term is 0 at $\tau = 1$ and approaches $\frac{1}{2}(G^2 + \sigma^2)$ as $\tau \to \infty$, quantifying the classic trade-off between fewer communications and increased drift.

5. **Noise floor and tuning guidelines:** The bracketed expression in Theorem 1 is a $T$-independent error floor. Once the $O(1/T)$ term becomes negligible as the number of rounds grows, additional rounds do not improve the bound unless one (a) increases $M$, (b) reduces heterogeneity (e.g., via smarter client selection that lowers $\zeta^2$), or (c) decreases $\tau$ to curb local drift.

## 5 MAIN EXPERIMENTS

**Experiment Setup.** In this section, we aim to investigate the performance of FEDAGENT under a uniform client distribution, which is independent of the aforementioned three types of client heterogeneities. We partitioned the whole dataset (WebShop or ALFWorld) into 100 clients. Each client has 100 task instructions and there is a potential overlap between clients. 2 clients are randomly selected each round. Each client is trained for 3 epochs per round, with a total of 70 rounds and 210 epochs overall. For each epoch, 64 tasks are sampled iteratively with replacement from local data.

As for FEDAGENT, we adopt GRPO (Shao et al., 2024) for policy optimization. Then, following the literature in federated learning (Liu et al., 2024), we select two typical baselines: **Centralized Agent Training** and **Local Agent Training**. Centralized Agent Training uses the full dataset (*i.e.*, 64 tasks are sampled iteratively from the whole dataset each epoch), while Local Agent Training uses only a specific client's dataset (we selected client index 21, 42, or 84 as the baselines). Both of them run for the same total epochs as FEDAGENT and also adopt GRPO for policy optimization.

**Result Analysis.** As shown in Table 1, **FEDAGENT consistently outperforms Local Agent Training and achieves comparable performance to Centralized Agent Training**. For instance, on ALFWorld using `Qwen2.5-1.5B-Instruct`, FEDAGENT achieves a 61.7% success rate compared to local training variants that range from 47.7% to 57.0%, while nearly matching the centralized training performance of 57.8%. This pattern is consistently observed across different model scales (1.5B, 3B, 7B), model architectures (qwen and `llama`), and client indexes (21, 42, and 84). Similarly, on the WebShop benchmark, FEDAGENT maintains this advantage with `Qwen2.5-7B-Instruct` achieving 68.9% success rate versus local training with different indexes ranging from 33.6% to 49.2%, while remaining competitive with centralized training at 64.7%. These results demonstrate the advantage of FEDAGENT in achieving competitive performance while preserving users' data privacy inherently.

| Method | ALFWorld | | | | | | | WebShop | |
|---|---|---|---|---|---|---|---|---|---|
| | Pick | Look | Clean | Heat | Cool | Pick2 | All | Score | Succ. |
| *Qwen2.5-1.5B-Instruct* | | | | | | | | | |
| Local (Client 21) | 42.9 | 25.0 | 38.5 | 37.5 | 14.3 | 14.3 | 29.7 | 69.9 | 57.0 |
| Local (Client 42) | 50.0 | 37.5 | **76.9** | 25.0 | 42.9 | 14.3 | 45.3 | 75.1 | 53.1 |
| Local (Client 84) | 50.0 | 37.5 | 46.2 | 25.0 | 28.6 | 0.0 | 34.4 | 72.7 | 47.7 |
| Centralized | $64.3_{\pm4.8}$ | $37.5_{\pm0.9}$ | $69.2_{\pm6.1}$ | $\mathbf{50.0}_{\pm2.2}$ | $42.9_{\pm3.8}$ | $28.6_{\pm0.4}$ | $51.6_{\pm3.0}$ | $79.9_{\pm4.7}$ | $57.8_{\pm5.7}$ |
| **FEDAGENT** | $\mathbf{80.0}_{\pm4.2}$ | $\mathbf{75.0}_{\pm1.7}$ | $53.8_{\pm4.3}$ | $37.5_{\pm1.3}$ | $\mathbf{83.3}_{\pm4.7}$ | $\mathbf{50.0}_{\pm1.0}$ | $\mathbf{64.1}_{\pm2.8}$ | $\mathbf{83.2}_{\pm4.5}$ | $\mathbf{61.7}_{\pm1.8}$ |
| *Qwen2.5-3B-Instruct* | | | | | | | | | |
| Local (Client 21) | 41.5 | 12.5 | 34.9 | **51.0** | 18.9 | 21.2 | 31.3 | 59.8 | 55.0 |
| Local (Client 42) | 46.5 | 37.5 | 24.4 | 15.0 | 33.7 | 33.3 | 28.2 | 61.3 | 59.3 |
| Local (Client 84) | 22.8 | 27.5 | 39.1 | 46.3 | 48.3 | 36.5 | 29.9 | 77.6 | 58.6 |
| Centralized | $94.1_{\pm0.9}$ | $\mathbf{80.0}_{\pm2.5}$ | $\mathbf{64.3}_{\pm1.4}$ | $42.9_{\pm2.6}$ | $50.0_{\pm2.7}$ | $22.2_{\pm5.2}$ | $62.5_{\pm4.2}$ | $70.0_{\pm1.5}$ | $53.9_{\pm2.8}$ |
| **FEDAGENT** | $\mathbf{95.5}_{\pm4.3}$ | $62.5_{\pm3.0}$ | $49.7_{\pm1.7}$ | $47.5_{\pm2.4}$ | $\mathbf{85.3}_{\pm3.6}$ | $\mathbf{45.1}_{\pm2.1}$ | $\mathbf{65.2}_{\pm3.9}$ | $\mathbf{85.5}_{\pm3.4}$ | $\mathbf{63.1}_{\pm3.1}$ |
| *Qwen2.5-7B-Instruct* | | | | | | | | | |
| Local (Client 21) | 35.5 | 25.0 | 61.0 | 25.9 | 35.8 | **45.2** | 38.4 | 70.9 | 49.2 |
| Local (Client 42) | 29.0 | 45.0 | 18.8 | 25.6 | 15.9 | 38.0 | 42.1 | 78.2 | 33.6 |
| Local (Client 84) | 34.7 | 47.5 | 44.4 | 51.3 | 40.1 | 21.8 | 35.7 | 60.6 | 39.3 |
| Centralized | $93.7_{\pm4.5}$ | $82.5_{\pm2.1}$ | $\mathbf{71.5}_{\pm3.3}$ | $47.9_{\pm3.7}$ | $63.2_{\pm3.8}$ | $31.9_{\pm1.0}$ | $73.3_{\pm4.0}$ | $78.8_{\pm2.8}$ | $64.7_{\pm1.6}$ |
| **FEDAGENT** | $\mathbf{94.5}_{\pm2.3}$ | $\mathbf{85.0}_{\pm4.1}$ | $56.0_{\pm0.8}$ | $\mathbf{62.5}_{\pm1.2}$ | $\mathbf{86.7}_{\pm2.9}$ | $42.8_{\pm3.4}$ | $\mathbf{75.5}_{\pm2.9}$ | $\mathbf{89.0}_{\pm4.1}$ | $\mathbf{68.9}_{\pm3.8}$ |
| *Llama-3.2-3B-Instruct* | | | | | | | | | |
| Local (Client 21) | 39.8 | 50.0 | 17.9 | 40.0 | 20.7 | 34.0 | 38.1 | 65.3 | 50.5 |
| Local (Client 42) | 18.2 | 55.0 | 41.9 | 34.3 | 41.0 | 25.0 | 35.0 | 67.0 | 51.0 |
| Local (Client 84) | 29.9 | 32.5 | 39.0 | 18.9 | 18.8 | **37.6** | 29.7 | 70.2 | 55.7 |
| Centralized | $72.4_{\pm4.6}$ | $\mathbf{62.5}_{\pm4.5}$ | $59.3_{\pm3.1}$ | $45.2_{\pm0.5}$ | $53.7_{\pm2.2}$ | $27.9_{\pm3.0}$ | $54.9_{\pm2.9}$ | $72.3_{\pm3.7}$ | $56.2_{\pm1.6}$ |
| **FEDAGENT** | $\mathbf{83.7}_{\pm1.7}$ | $57.5_{\pm6.0}$ | $\mathbf{60.6}_{\pm3.4}$ | $\mathbf{55.9}_{\pm0.9}$ | $\mathbf{65.3}_{\pm2.8}$ | $24.9_{\pm3.1}$ | $\mathbf{61.2}_{\pm3.3}$ | $\mathbf{74.4}_{\pm4.9}$ | $\mathbf{57.8}_{\pm3.2}$ |

Table 1: **Performance Comparison on ALFWorld and WebShop**. We report the averaged performance and the corresponding standard deviation for Centralized Training and FEDAGENT over three random seeds. For ALFWorld, the **Success Rate** (%) is reported for each subtask as well as for the overall dataset. For WebShop, both the **Task Score** (%) and the **Success Rate** (%) are reported.

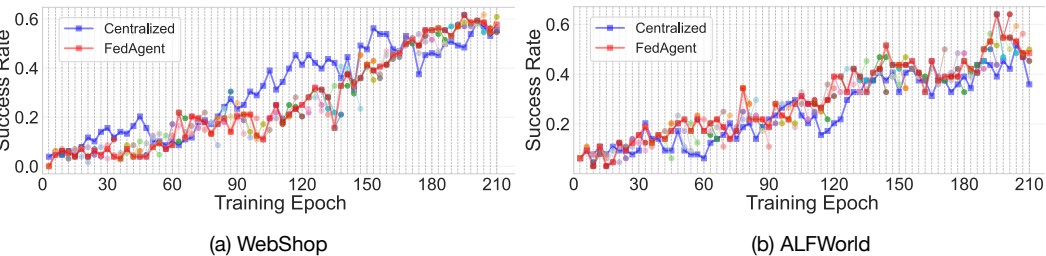

(a) WebShop
(b) ALFWorld

Figure 2: **Training Dynamics of FEDAGENT and Centralized Training**. Circle marks with different colors indicate the model performance after training on *specific selected clients each round*. The red line refers to the performance of the *aggregated models on server* throughout the training process.

Figure 2 shows the whole training dynamics of FEDAGENT and Centralized Agent Training with Qwen2.5-1.5B-Instruct on WebShop and ALFWorld datasets. **Both paradigms ultimately converge to similar success rates despite different training dynamics** ($\sim 0.6$ for WebShop, $\sim 0.5$ for ALFWorld). In WebShop (left), both approaches demonstrate steady monotonic improvement, with centralized training initially outperforming FEDAGENT until approximately epoch 120, after which both converge to similar success rates around 0.6. In contrast, ALFWorld (right) exhibits relatively more volatile training dynamics with frequent performance fluctuations for both methods, ultimately converging to success rates around 0.5. This further illustrates that FEDAGENT can achieve comparable performance with centralized training.

## 6 IMPACT OF DIFFERENT DECENTRALIZED SETTINGS

**Experiment Setup.** In this section, we aim to study the impact of different decentralized settings on FEDAGENT in FEDAGENTGYM by systematically varying three key hyperparameters across two

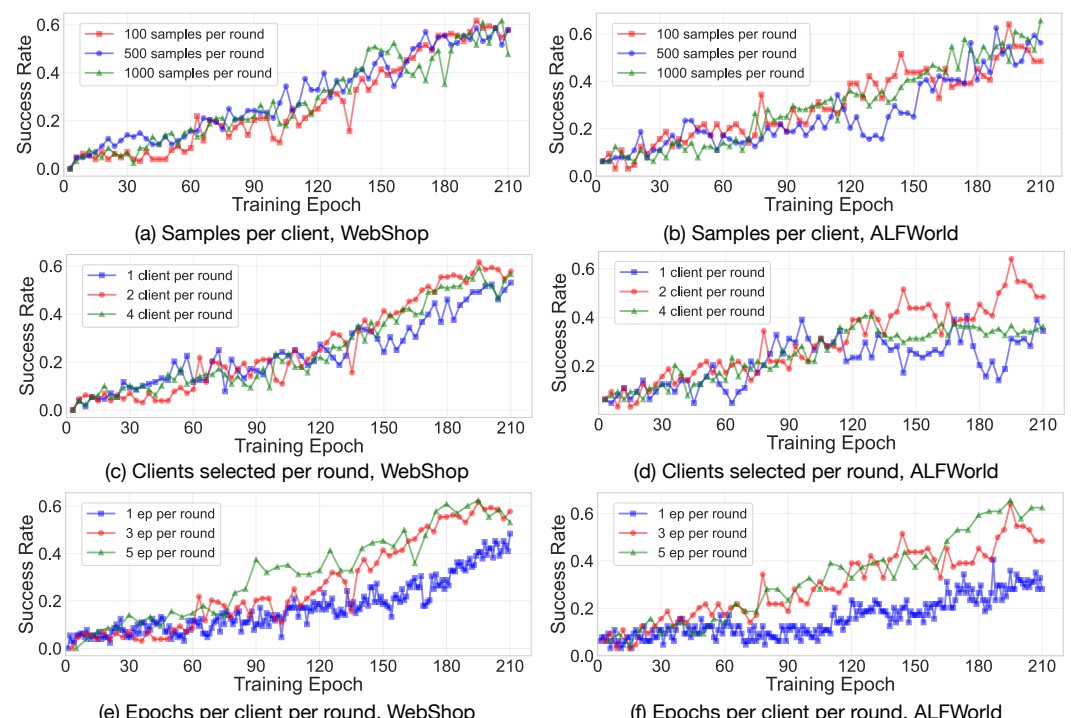

Figure 3: **Training Dynamics of FEDAGENT in Different Decentralized Settings**.

different datasets (WebShop and ALFWorld). We adopt `Qwen2.5-1.5B-Instruct` for all configurations. The experimental setup examines: (1) **samples per client**. We test $100$, $500$, and $1,000$ tasks per client to understand how task sampling scope affects FEDAGENT learning dynamics; (2) **clients selected per round**. We compare $1$, $2$, and $4$ participating clients each round to analyze the effect of federation scale on performance; and (3) **epochs per client per round**. We evaluate $1$, $3$, and $5$ local training epochs to determine the optimal number of local computations before aggregation. Since we keep the total number of epochs the same at $210$ for all configurations, $1$, $3$, and $5$ local training epochs correspond to $210$, $70$, and $42$ total rounds, respectively.

**Result Analysis.** The results in Figure 3 demonstrate that **FEDAGENT exhibits distinct sensitivity patterns towards decentralized settings, depending on the specific hyperparameter and dataset**. First, it shows notable sensitivity to the number of epochs per client per round. Moving from $1$ to $5$ epochs per round leads to significant performance gains, especially after around $100$ training epochs, highlighting that shallow local updates are insufficient to unlock the full potential of FEDAGENT. On ALFWorld, FEDAGENT is also sensitive to the number of clients selected per round, with $2$ clients per round outperforming $1$ or $4$, suggesting that too few or too many clients could hinder convergence. By contrast, FEDAGENT appears insensitive to the number of samples per client, as performance curves largely overlap across $100$, $500$, and $1,000$ samples per round, suggesting that the task sampling scope for one client beyond a certain threshold may not be the limiting factor. Our studies offer valuable insights on the practical deployment of FEDAGENT and also suggest that **optimal federated agent learning configurations are environment-dependent**.

## 7  IMPACT OF HETEROGENEITY CHALLENGES

**Experiment Setup.** In this section, we aim to study the impact of different heterogeneity challenges on FEDAGENT in FEDAGENTGYM. As shown in Appendix B.2, we can leverage our proposed client partitioning strategies PREFERENCEPARTITION, COVERAGEPARTITION, and HARDNESSPARTITION to precisely control the extent of one form of heterogeneity (Preference, Coverage, or Hardness Heterogeneity) across clients with a hyperparameter $\omega$, $\xi$, or $\xi'$. We keep the number of total epochs as $210$ and the number of all clients as $100$, which are consistent with the main experiments. We adopt `Qwen2.5-1.5B-Instruct` in the experiments.

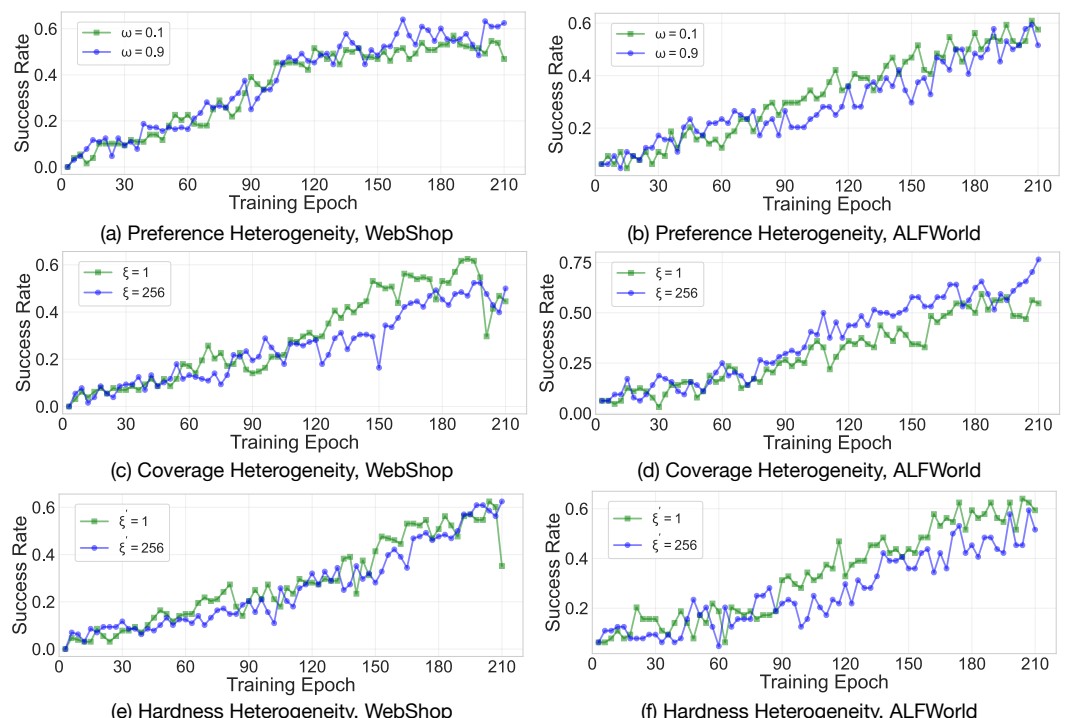

Figure 4: **Training Dynamics of FEDAGENT in Different Heterogeneity Challenges**.

**Result Analysis.**    As shown in Figure 4, **FEDAGENT shows high robustness against the three heterogeneity challenges**. Across all scenarios, preference heterogeneity (panels a,b), coverage heterogeneity (panels c,d), and hardness heterogeneity (panels e,f), even when comparing low heterogeneity settings ($\omega = 0.1$, $\xi = 256$, $\xi' = 256$) against high heterogeneity settings ($\omega = 0.9$, $\xi = 1$, $\xi' = 1$), FEDAGENT consistently achieves strong success rates that steadily improve throughout training, The learning curves show that FEDAGENT maintains stable convergence behavior in both WebShop and ALFWorld environments regardless of heterogeneity intensity, with success rates generally reaching 0.5-0.6 by the end of training. Crucially, the performance degradation is minimal even under extreme heterogeneity conditions, indicating that **FEDAGENT has great potential to handle real-world scenarios across the full spectrum of heterogeneity challenges**.

## 8    RELATED WORK

RL has been instrumental in empowering LLM agents to function effectively in dynamic and open-ended environments. Initial studies leveraged traditional RL approaches like DQN (Mnih et al., 2015) for training LLM agents in text-based gaming environments (Narasimhan et al., 2015). Subsequent research began incorporating value-based techniques across broader agent applications such as Android device manipulation (Rawles et al., 2023) and embodied environments like ALFWorld (Shridhar et al., 2020). Contemporary methods have expanded RL training to encompass sophisticated web-based and application-specific tasks (Zhou et al., 2024; Putta et al., 2024). In previous works, real-world task queries and trajectories have been essential for training AI agents in practical applications. However, they are becoming increasingly difficult to acquire due to privacy concerns. Our work makes an initial effort to explore training AI agents without compromising user data privacy.

## 9    CONCLUSION

In this work, we explored FEDAGENT (Federated Agent Reinforcement Learning), a new collaborative paradigm to train AI agent, particularly LLMs, across distributed clients, and built FEDAGENTGYM, the first decentralized agent learning environment. Extensive theoretical and empirical studies demonstrate that FEDAGENT can achieve performance on par with centralized training and maintain strong robustness to heterogeneities. Our work validates the feasibility of training AI agents while protecting user data privacy and charts new research directions in agent learning.

## ETHICS STATEMENT

This research on federated agent reinforcement learning aims to address the critical privacy concerns in AI agent training by developing decentralized paradigms that eliminate centralized data collection, ensuring all user data remains distributed across local clients throughout the training process.

## REPRODUCIBILITY STATEMENT

The link to the code repository is available here. The proof of Theorem 1 is in Appendix C.

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

# Content of Appendix

## A    LLM USAGE DISCLOSURE

We hereby disclose that Large Language Models (LLMs) are utilized solely for the purposes of grammar correction and textual refinement.

# B  MORE DETAILS OF HETEROGENEITY CHALLENGES

## B.1  PSEUDO CODE FOR CLIENT PARTITIONING STRATEGIES

---

**Algorithm 2** PREFERENCEPARTITION

---

**Require:** Category pools $\{\mathcal{I}_c\}_{c=1}^C$ with sizes $n_c$; total clients $K$; per-client set size $L$; jitter $\omega$
**Ensure:** Client datasets $X_1, \ldots, X_K$ with $|X_k| = L$
1:  $p_c \leftarrow n_c / \sum_{j=1}^C n_j; \quad \ell_c \leftarrow \log \frac{p_c}{1-p_c}$ ▷ global mix + logit anchors
2:  **for** $k = 1$ to $K$ **do**
3:      $z_c \sim \mathcal{N}(\ell_c, \omega^2)$ for $c = 1 \ldots C; \quad q_c \leftarrow \exp(z_c) / \sum_j \exp(z_j)$ ▷ larger $\omega \Rightarrow$ higher variance
4:      $(a_1, \ldots, a_C) \sim \text{Multinomial}(L; q_1, \ldots, q_C)$ ▷ category counts for client $k$
5:      **if** any $a_c > n_c$ **then** set $a_c \leftarrow \min(a_c, n_c)$ and redistribute leftover by $q$ to classes with spare capacity ▷ capacity fix within a set
6:      $X_k \leftarrow \bigcup_{c=1}^C \text{SAMPLEWITHOUTREPLACEMENT}(\mathcal{I}_c, a_c)$
7:  **end for**
8:  **return** $\{X_k\}_{k=1}^K$

---

---

**Algorithm 3** COVERAGEPARTITION

---

**Require:** total items $N$ (indexed 1:$N$); total clients $K$; per-client bounds $(L_{\min}, L_{\text{avg}}, L_{\max})$ with $L_{\min} \leq L_{\text{avg}} \leq L_{\max}$; dispersion $\xi$; desired average replicas per item $r$
**Ensure:** Client datasets $X_1, \ldots, X_K$
1:  $T \leftarrow \lfloor rN \rfloor$ ▷ total assignments (sum of all $|X_k|$); keeps global overlap fixed
2:  **assert** $KL_{\min} \leq T \leq KL_{\max}$ ▷ feasibility under per-client bounds
3:  $\mu \leftarrow (L_{\text{avg}} - L_{\min}) / (L_{\max} - L_{\min}); \quad \alpha \leftarrow \mu\xi, \ \beta \leftarrow (1 - \mu)\xi$ ▷ Beta params with mean fixed at $L_{\text{avg}}$
4:  Sample $x_k \sim \text{Beta}(\alpha, \beta)$ for $k = 1 \ldots K$ ▷ larger $\xi \Rightarrow$ lower variance (sizes closer to $L_{\text{avg}}$)
5:  $u_k \leftarrow L_{\min} + x_k(L_{\max} - L_{\min}); \quad u_k \leftarrow u_k \cdot \dfrac{T}{\sum_j u_j}$ ▷ shape then renormalize to sum $T$
6:  $n_k \leftarrow \text{ROUNDTOSUM}(u, T, [L_{\min}, L_{\max}])$ ▷ largest remainder with clipping to $[L_{\min}, L_{\max}]$
7:  $m \leftarrow \lfloor r \rfloor, \quad M \leftarrow \lceil r \rceil, \quad H \leftarrow T - mN$
8:  Set $q_i \leftarrow M$ for any $H$ items; $q_i \leftarrow m$ otherwise
9:  Initialize $X_k \leftarrow \emptyset$, $\text{rem}_k \leftarrow n_k$ for all $k$
10: **for** $i = 1$ to $N$ **do** ▷ weighted, no-replacement placement across clients
11:     $\mathcal{A} \leftarrow \{k : \text{rem}_k > 0\}$; choose $q_i$ distinct $k \in \mathcal{A}$ with $\Pr(k) \propto \text{rem}_k$
12:     Add item $i$ to each chosen $X_k$ and decrement the corresponding $\text{rem}_k$
13: **end for**
14: **return** $\{X_k\}_{k=1}^K$

---

---

**Algorithm 4** HARDNESSPARTITION

---

**Require:** total items $N$ (indexed 1:$N$); disjoint index sets $\mathcal{S}$ (successful) and $\mathcal{U}$ (unsuccessful) with $\mathcal{S} \cup \mathcal{U} = \{1{:}N\}$; total clients $K$; per-client set size $L$; Hyperparameters for COVERAGEPARTITION: bounds $(\ell, c, h)$ with $h \leq L$, dispersion $\xi'$, overlap $r$
**Ensure:** client datasets $X_1, \ldots, X_K$ with $|X_k| = L$
1:  $\{Y_k\}_{k=1}^K \leftarrow \text{COVERAGEPARTITION}(|\mathcal{S}|, K, (\ell, c, h), \xi', r)$ ▷ larger $\xi' \Rightarrow$ lower variance
2:  **for** $k = 1$ to $K$ **do**
3:      $m_k \leftarrow L - |Y_k|; \quad F_k \leftarrow \text{SAMPLEWITHOUTREPLACEMENT}(\mathcal{U}, m_k)$
4:      $X_k \leftarrow Y_k \cup F_k$
5:  **end for**
6:  **return** $\{X_k\}_{k=1}^K$

---

## B.2 CLIENT DISTRIBUTIONS UNDER PARTITIONING STRATEGIES

### B.2.1 PREFERENCE HETEROGENEITY

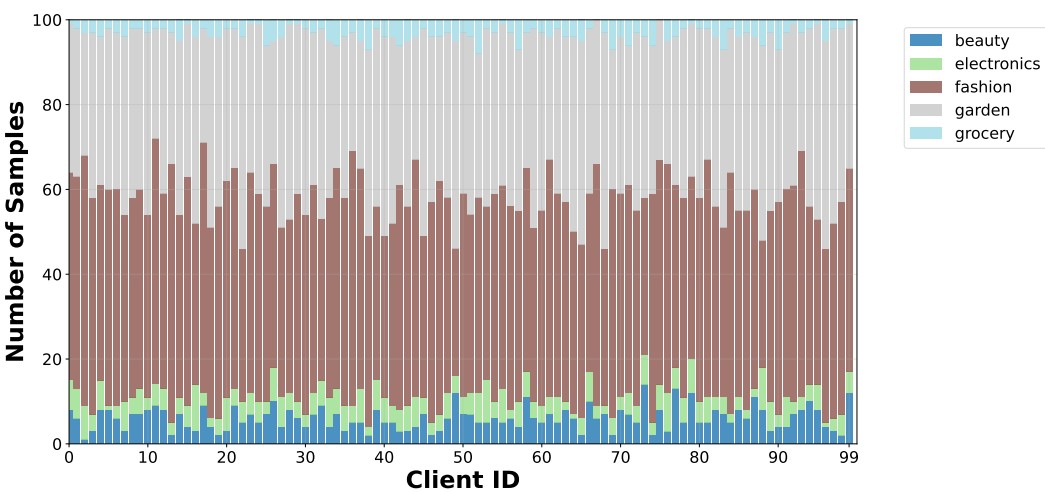

Figure 5: **Client Distribution under Preference Heterogeneity (WebShop, $\omega = 0.1$).**

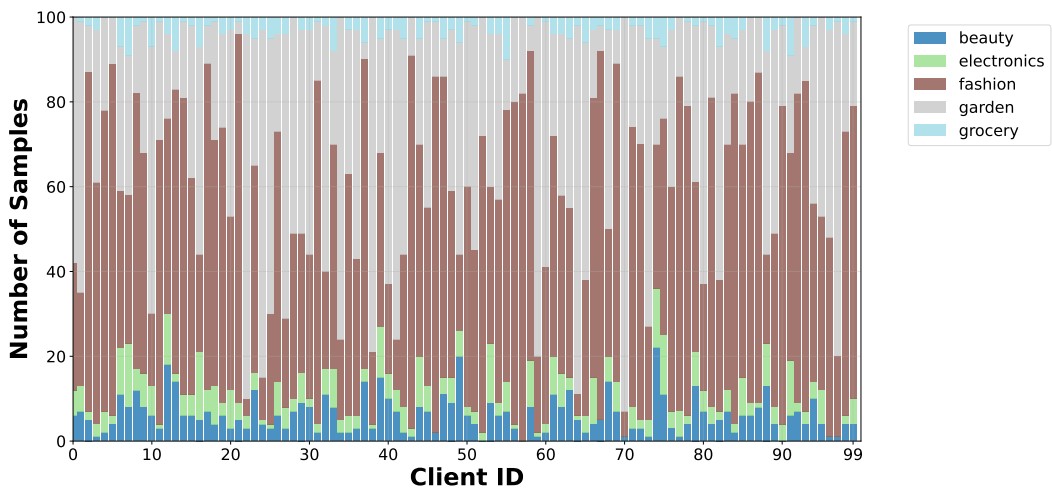

Figure 6: **Client Distribution under Preference Heterogeneity (WebShop, $\omega = 0.9$).**

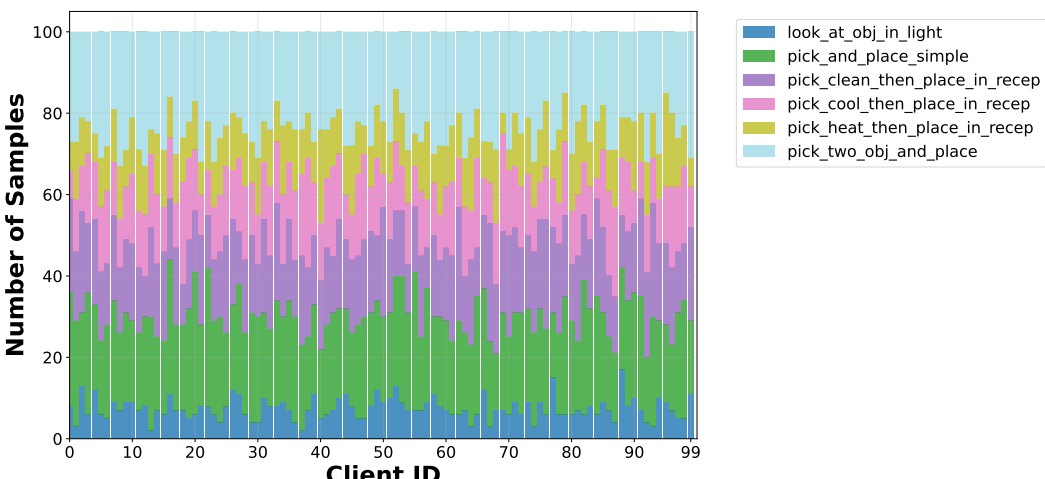

Figure 7: **Client Distribution under Preference Heterogeneity (ALFWorld, $\omega = 0.1$).**

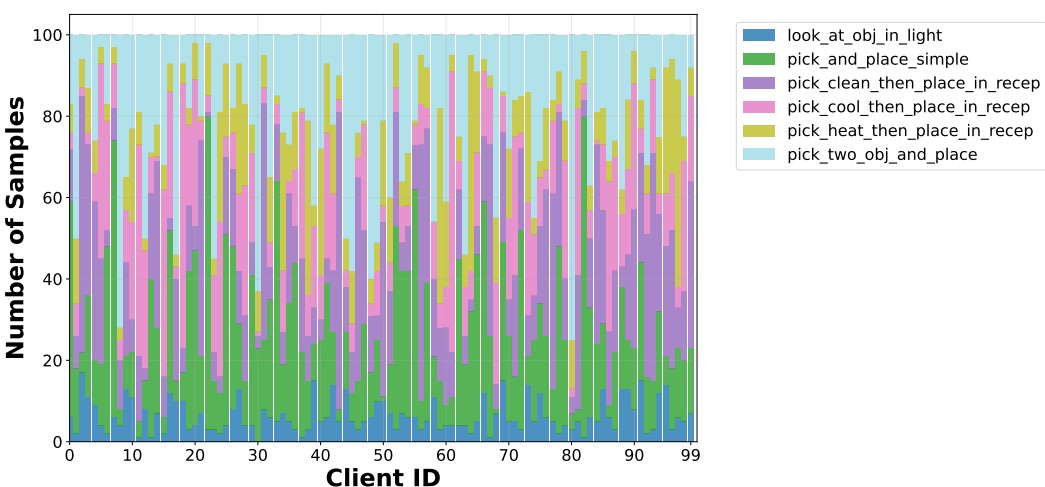

Figure 8: **Client Distribution under Preference Heterogeneity (ALFWorld, $\omega = 0.9$).**

### B.2.2 COVERAGE HETEROGENEITY

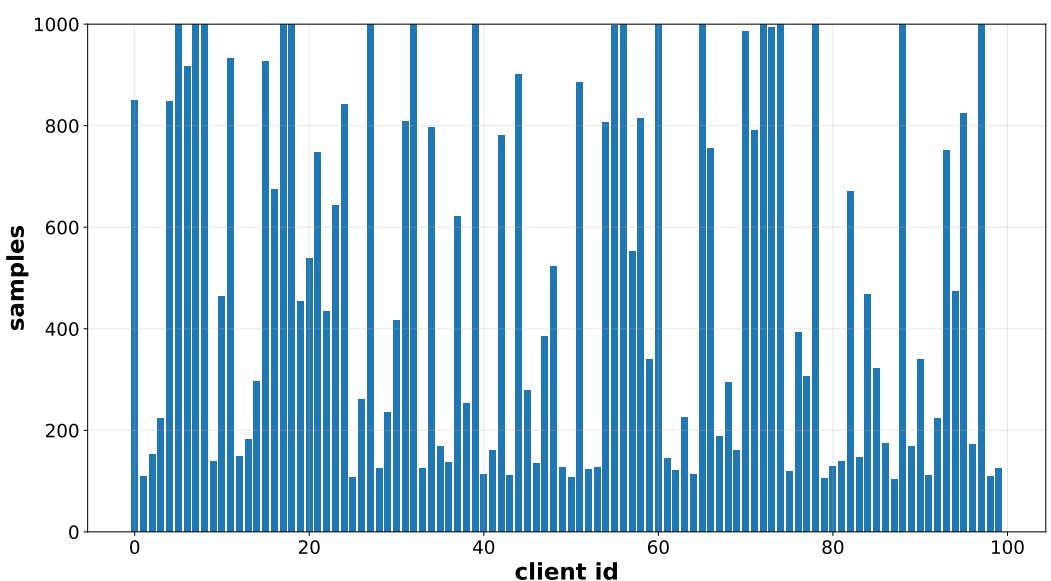

Figure 9: **Client Distribution under Coverage Heterogeneity (WebShop, $\xi = 1$).**

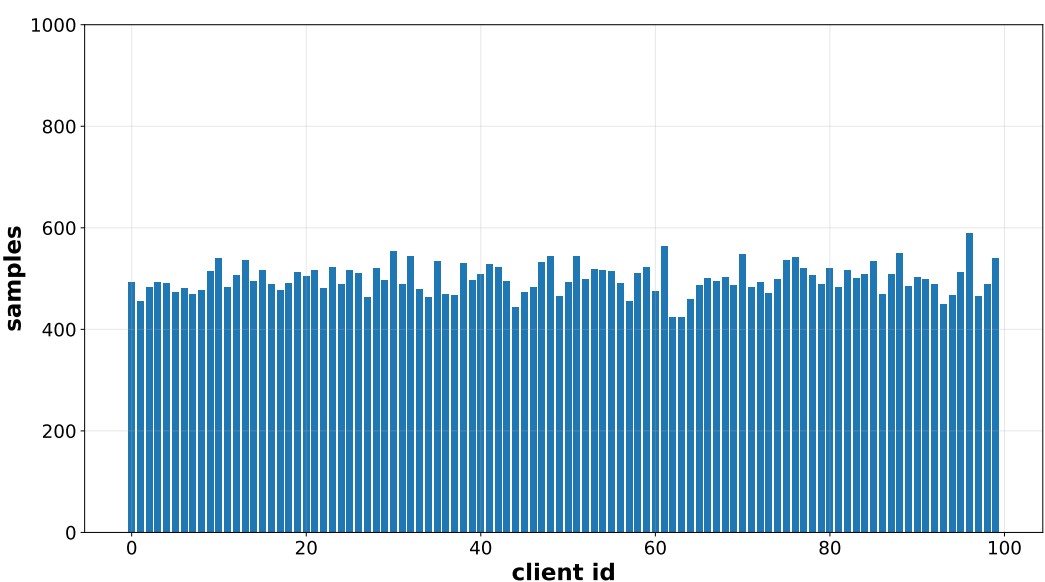

Figure 10: **Client Distribution under Coverage Heterogeneity (WebShop, $\xi = 256$).**

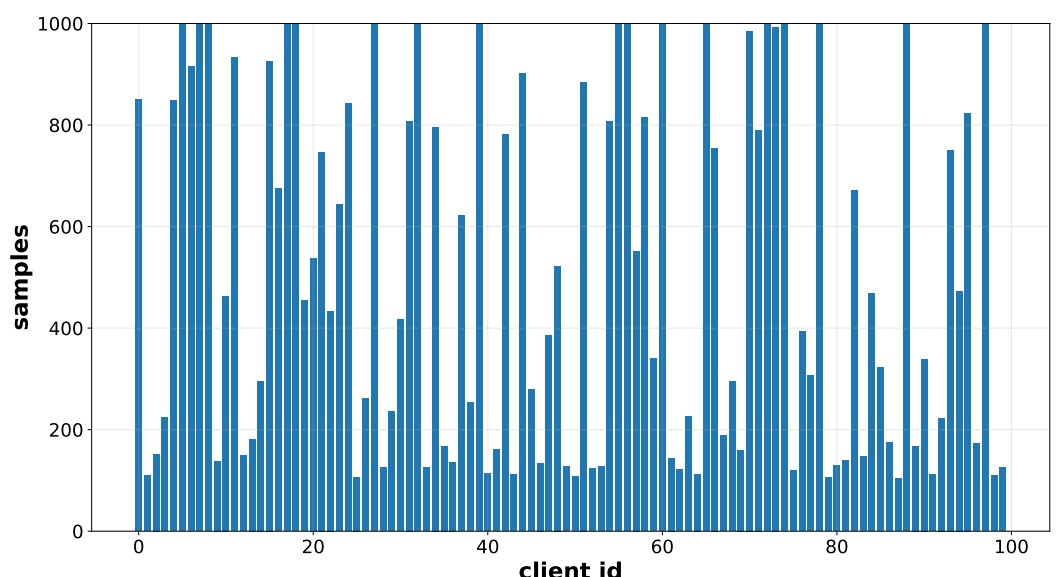

Figure 11: **Client Distribution under Coverage Heterogeneity (ALFWorld, $\xi = 1$).**

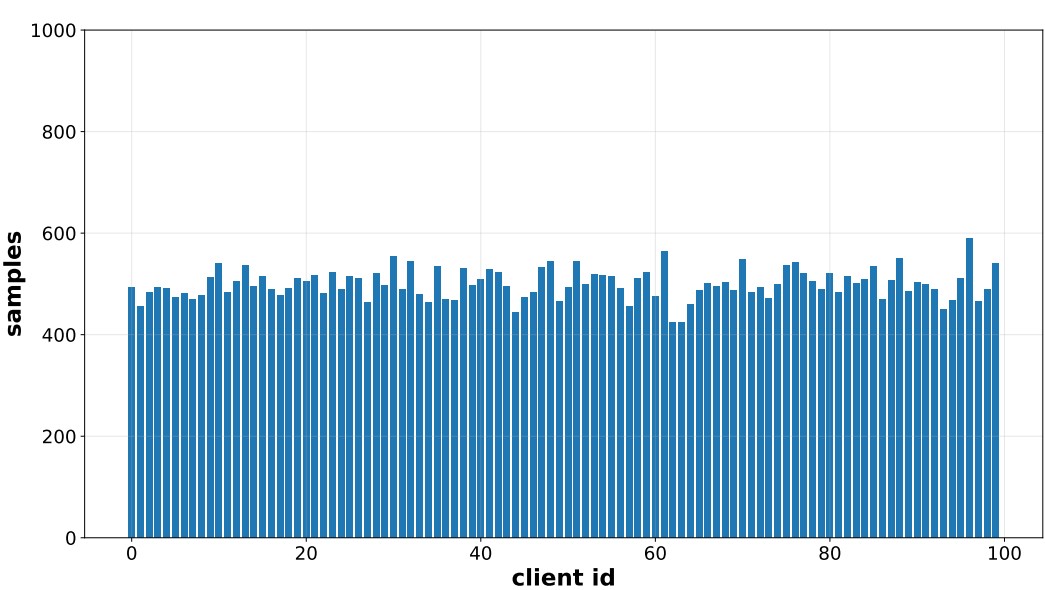

Figure 12: **Client Distribution under Coverage Heterogeneity (ALFWorld, $\xi = 256$).**

### B.2.3  HARDNESS HETEROGENEITY

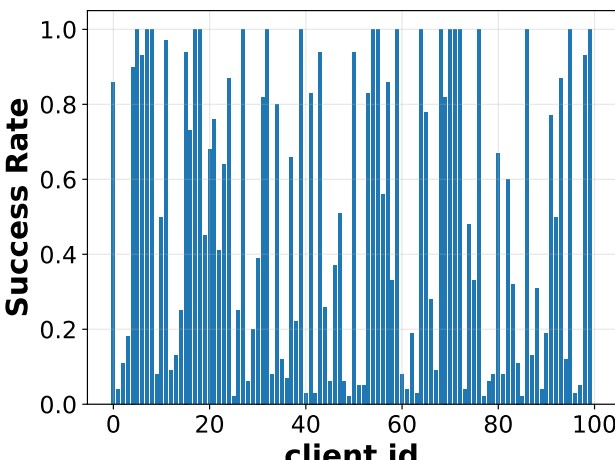

Figure 13: **Client Distribution under Coverage Heterogeneity (WebShop, $\xi' = 1$).**

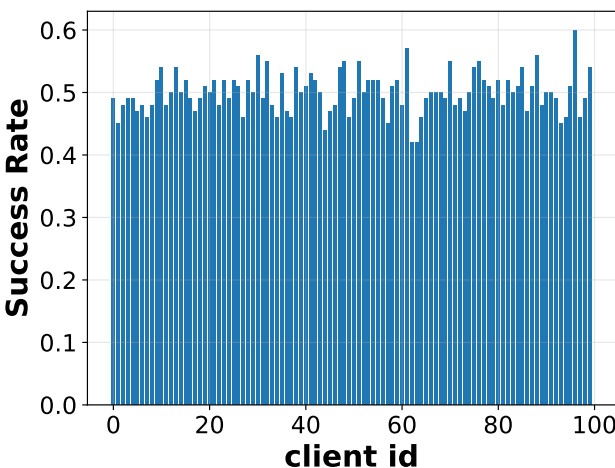

Figure 14: **Client Distribution under Coverage Heterogeneity (WebShop, $\xi' = 256$).**

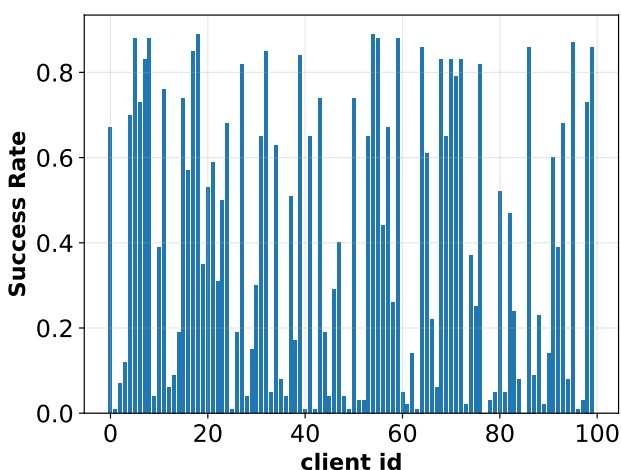

Figure 15: **Client Distribution under Coverage Heterogeneity (ALFWorld, $\xi' = 1$).**

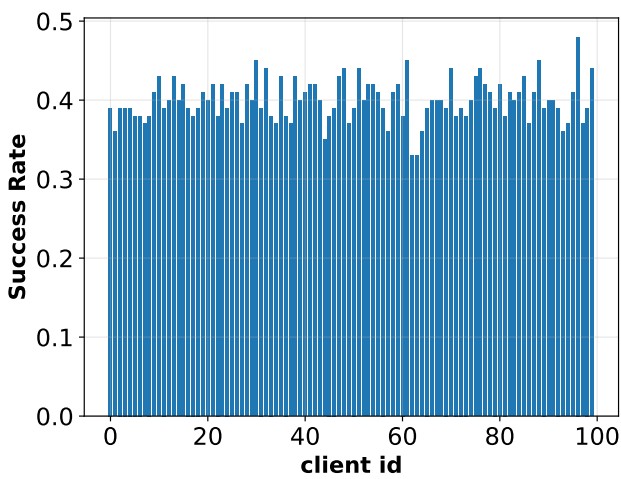

Figure 16: **Client Distribution under Coverage Heterogeneity (ALFWorld, $\xi' = 256$).**

## C    PROOF OF THE CONVERGENCE

**Assumption 1** (*L*-smooth gradients). *For all $\theta, \theta' \in \mathbb{R}^d$ and $k \in [K]$, the client objectives have L-Lipschitz gradients:*

$$\|\nabla J_k(\theta) - \nabla J_k(\theta')\| \leq L\|\theta - \theta'\|.$$

**Assumption 2** (*G*-bounded gradients). *For all $\theta \in \mathbb{R}^d$ and $k \in [K]$, the full gradients are bounded:*

$$\|\nabla J_k(\theta)\| \leq G.$$

**Assumption 3** (*σ*-bounded variance). *For all $\theta \in \mathbb{R}^d$ and $k \in [K]$, the stochastic gradient estimator has bounded variance:*

$$\mathbb{E}\big[\|\nabla J_k(\theta; B) - \nabla J_k(\theta)\|^2\big] \leq \sigma^2,$$

*where $\nabla J_k(\theta; B)$ denotes the mini-batch gradient.*

**Assumption 4** (Polyak–Łojasiewicz (PL) condition). *The global objective satisfies, for some $\mu > 0$ and $\theta^\star = \arg\max_\theta J(\theta)$,*

$$2\mu\big(J(\theta^\star) - J(\theta)\big) \ \leq \ \|\nabla J(\theta)\|^2, \qquad \forall\, \theta \in \mathbb{R}^d.$$

**Assumption 5** (Bounded client heterogeneity). *There exists $\zeta^2$ such that for all $\theta$,*

$$\frac{1}{K}\sum_{k=0}^{K-1}\big\|\nabla J_k(\theta) - \nabla J(\theta)\big\|^2 \ \leq \ \zeta^2, \qquad \text{where } \nabla J(\theta) = \frac{1}{K}\sum_{k=0}^{K-1}\nabla J_k(\theta).$$

**Remark.** Assumptions 1-3 are standard in stochastic optimization literature. As for Assumption 4 (the PL condition), in practice, policy-gradient methods that constrain update size, such as trust-region approaches or proximal policy methods, yield smoother policy updates, making the PL assumption more tenable. Recent works have likewise employed PL-type conditions to obtain convergence guarantees for non-convex reinforcement learning objectives (Bhandari & Russo, 2024; Karimi et al., 2016; Yuan et al., 2022), supporting our adoption of this assumption. Assumption 5 is a common "bounded heterogeneity" condition used to control client drift in federated learning analyses (Li et al., 2020; Karimireddy et al., 2020; Stich, 2018; Khaled et al., 2020; Woodworth et al., 2020).

**Theorem 1** (Convergence of FEDAGENT). *Under Assumptions 1–5, suppose that at each communication round $t$ the server uniformly samples without replacement a subset $S_t \subset [K]$ of size $M \leq K$ and aggregates only those clients' updates: $\theta_{t+1} = \theta_t + \frac{1}{M}\sum_{k \in S_t}\Delta\theta_{k,t}$, with the same local inner loop and notation as in alg:fedagent. Let each selected client perform $\tau$ local steps with stepsize $\eta$. Choose the stepsize $\eta = \frac{1}{L\tau}$ and let $\tilde{\theta}$ be a uniform random iterate drawn from $\{\theta_t\}_{t=0}^{T-1}$. Then*

$$\mathbb{E}\Big[J(\theta^\star) - J(\tilde{\theta})\Big] \ \leq \ \frac{L}{\mu T}\big(J(\theta^\star) - J(\theta_0)\big) + \frac{1}{2\mu}\left[\frac{G^2 + \sigma^2}{M} + \frac{2(K-M)}{M(K-1)}\zeta^2 + \frac{(\tau-1)^2}{2\tau^2}\left(G^2 + \sigma^2\right)\right].$$

*In particular, the convergence rate remains $O(1/T)$. The "noise floor" comprises: (i) an $O(1/M)$ local stochastic term, (ii) an $O\big(\frac{K-M}{M(K-1)}\big)$ heterogeneity term due to client sampling, and (iii) an $O\big(\frac{(\tau-1)^2}{\tau^2}\big)$ local-drift term that vanishes when $\tau = 1$.*

### C.1    PROOF SKETCH

*Proof sketch.* Our proof generally follows the proof of Theorem 4.1 in (Fan et al., 2025), with key modifications on second-moment bound for the aggregated update and local-drift term.

Let $u_{k,t} := \sum_{i=0}^{\tau-1} g_{k,t,i}$ be client $k$'s aggregate local stochastic gradients in round $t$, and $\bar{g}_{k,t} := \frac{1}{\tau}\sum_{i=0}^{\tau-1} g_{k,t,i}$. Define the round-average $\bar{g}_t := \frac{1}{M}\sum_{k \in S_t}\bar{g}_{k,t}$, so the server update is $\Delta_t = \theta_{t+1} - \theta_t = \eta\tau\bar{g}_t$.

**(1) One-step descent.** By $L$-smoothness of $J$,

$$\mathbb{E}[J(\theta_{t+1}) \mid \theta_t] \ \geq \ J(\theta_t) + \eta\tau\Big(1 - \frac{L\eta\tau}{2}\Big)\|\nabla J(\theta_t)\|^2 - \frac{L}{2}\eta^2\tau^2\,\mathbb{E}\|\bar{g}_t - \nabla J(\theta_t)\|^2 \qquad (4)$$

**(2) Variance–bias decomposition with Finite-Population Correction (FPC).** Decompose

$$\bar{g}_t - \nabla J(\theta_t) = \underbrace{\left( \frac{1}{M} \sum_{k \in S_t} \nabla J_k(\theta_t) - \nabla J(\theta_t) \right)}_{\text{client sampling}} + \underbrace{\frac{1}{M} \sum_{k \in S_t} \left( \bar{g}_{k,t} - \mathbb{E}\bar{g}_{k,t} \right)}_{\text{local stochastic noise}} + \underbrace{\frac{1}{M} \sum_{k \in S_t} b_{k,t}}_{\text{local drift}},$$

where $b_{k,t} := \mathbb{E}[\bar{g}_{k,t} \mid \theta_t] - \nabla J_k(\theta_t)$. The three terms are bounded as follows:

$$\mathbb{E}\left\| \frac{1}{M} \sum_{k \in S_t} \nabla J_k(\theta_t) - \nabla J(\theta_t) \right\|^2 = \frac{(K-M)}{M(K-1)} \cdot \frac{1}{K} \sum_{k=1}^{K} \left\| \nabla J_k(\theta_t) - \nabla J(\theta_t) \right\|^2$$

$$\leq \frac{2(K-M)}{M(K-1)} \zeta^2, \quad \text{(FPC)}$$

$$\mathbb{E}\left\| \frac{1}{M} \sum_{k \in S_t} \left( \bar{g}_{k,t} - \mathbb{E}\bar{g}_{k,t} \right) \right\|^2 \leq \frac{G^2 + \sigma^2}{M}, \quad \text{(local noise)}$$

$$b_{k,t} = \frac{1}{\tau} \sum_{i=0}^{\tau-1} \left( \nabla J_k(\theta_{k,i}) - \nabla J_k(\theta_t) \right), \quad \mathbb{E}\|b_{k,t}\|^2 \leq \frac{L^2 \eta^2 (\tau-1)^2}{2} (G^2 + \sigma^2),$$

$$\Rightarrow \quad \mathbb{E}\left\| \frac{1}{M} \sum_{k \in S_t} b_{k,t} \right\|^2 \leq \frac{L^2 \eta^2 (\tau-1)^2}{2} (G^2 + \sigma^2). \quad \text{(local drift)}$$

Combining,

$$\mathbb{E}\left\| \bar{g}_t - \nabla J(\theta_t) \right\|^2 \leq \frac{G^2 + \sigma^2}{M} + \frac{2(K-M)}{M(K-1)} \zeta^2 + \frac{L^2 \eta^2 (\tau-1)^2}{2} (G^2 + \sigma^2). \tag{5}$$

**(3) PL and averaging.** Let $\delta_t := \mathbb{E}[J(\theta^\star) - J(\theta_t)]$. Applying the PL condition $\|\nabla J(\theta_t)\|^2 \geq 2\mu\, \delta_t$ in Equation (4) yields the linear recursion

$$\delta_{t+1} \leq \left( 1 - 2\mu\eta\tau \left( 1 - \frac{L\eta\tau}{2} \right) \right) \delta_t + \frac{L}{2} \eta^2 \tau^2 \left( \frac{G^2 + \sigma^2}{M} + \frac{2(K-M)}{M(K-1)} \zeta^2 + \frac{L^2 \eta^2 (\tau-1)^2}{2} (G^2 + \sigma^2) \right),$$

where, in forming Equation (5), we control the sampling-drift mixed term via Young's inequality $2\langle X, Y \rangle \leq \|X\|^2 + \|Y\|^2$ (thus inflating the sampling and drift pieces by a factor of 2). With $\eta = \frac{1}{L\tau}$ the contraction becomes $1 - \mu/L$, and the drift contribution simplifies to $\frac{(\tau-1)^2}{2\tau^2}(G^2 + \sigma^2)$. Unrolling the recursion and averaging the gaps gives

$$\frac{1}{T} \sum_{t=0}^{T-1} \delta_t \leq \frac{L}{\mu T} \delta_0 + \frac{1}{2\mu} \left( \frac{G^2 + \sigma^2}{M} + \frac{2(K-M)}{M(K-1)} \zeta^2 + \frac{(\tau-1)^2}{2\tau^2} (G^2 + \sigma^2) \right).$$

Finally, let $\tilde{\theta}$ be drawn uniformly at random from $\{\theta_t\}_{t=0}^{T-1}$; then $\mathbb{E}\left[ J(\theta^\star) - J(\tilde{\theta}) \right] = \frac{1}{T} \sum_{t=0}^{T-1} \delta_t$, which yields the stated bound.

$\square$

## C.2   A MORE DETAILED PROOF

*Proof.* We give a more detailed proof as follows. Our proof generally follows the proof of Theorem 4.1 in (Fan et al., 2025). Throughout, write $\mathbb{E}_t[\cdot] := \mathbb{E}[\cdot \mid \theta_t]$. Let each selected client $k \in S_t$ perform $\tau$ local stochastic policy-gradient steps with per-step gradients $g_{k,t,i}$, $i = 0, \ldots, \tau-1$,

$$\theta_{k,t,0} = \theta_t, \quad \theta_{k,t,i+1} = \theta_{k,t,i} + \eta\, g_{k,t,i}, \quad \mathbb{E}[g_{k,t,i} \mid \theta_{k,t,i}] = \nabla J_k(\theta_{k,t,i}), \quad \mathbb{E}\|g_{k,t,i}\|^2 \leq G^2 + \sigma^2.$$

Define the client's round aggregates $u_{k,t} := \sum_{i=0}^{\tau-1} g_{k,t,i}$ and $\bar{g}_{k,t} := \frac{1}{\tau} u_{k,t}$, and the server's round average

$$\bar{g}_t := \frac{1}{M} \sum_{k \in S_t} \bar{g}_{k,t}, \qquad \Delta_t := \theta_{t+1} - \theta_t = \eta\tau\, \bar{g}_t.$$

We analyze $J(\theta_{t+1})$ by Assumption 1 $L$-smoothness of $J$:

$$J(\theta_{t+1}) \geq J(\theta_t) + \left\langle \nabla J(\theta_t), \Delta_t \right\rangle - \frac{L}{2} \|\Delta_t\|^2 \tag{6}$$

**A. One-step progress.** Let $g_t^\star := \nabla J(\theta_t)$ and $e_t := \bar{g}_t - g_t^\star$. With $\eta = \frac{1}{L\tau}$ we have $\Delta_t = \frac{1}{L}(g_t^\star + e_t)$ and thus

$$
\begin{aligned}
\langle g_t^\star, \Delta_t \rangle - \frac{L}{2}\|\Delta_t\|^2 &= \frac{1}{L}\left(\|g_t^\star\|^2 + \langle g_t^\star, e_t \rangle\right) - \frac{1}{2L}\|g_t^\star + e_t\|^2 \\
&= \frac{1}{L}\left(\|g_t^\star\|^2 + \langle g_t^\star, e_t \rangle\right) - \frac{1}{2L}\left(\|g_t^\star\|^2 + 2\langle g_t^\star, e_t \rangle + \|e_t\|^2\right) \\
&= \frac{1}{2L}\|g_t^\star\|^2 - \frac{1}{2L}\|e_t\|^2.
\end{aligned}
\tag{7}
$$

Plugging equation 7 into equation 6 and taking $\mathbb{E}_t[\cdot]$,

$$
\mathbb{E}_t[J(\theta_{t+1})] \geq J(\theta_t) + \frac{1}{2L}\|g_t^\star\|^2 - \frac{1}{2L}\mathbb{E}_t\|e_t\|^2
\tag{8}
$$

Thus the entire task reduces to bounding $\mathbb{E}_t\|e_t\|^2$.

**B. Variance–bias decomposition of $\mathbb{E}_t\|e_t\|^2$.** We decompose $e_t$ into three parts:

$$
e_t = \underbrace{\left(\frac{1}{M}\sum_{k\in S_t}\nabla J_k(\theta_t) - \nabla J(\theta_t)\right)}_{\text{client sampling}} + \underbrace{\frac{1}{M}\sum_{k\in S_t}\left(\bar{g}_{k,t} - \mathbb{E}_t\bar{g}_{k,t}\right)}_{\text{local stochastic noise}} + \underbrace{\frac{1}{M}\sum_{k\in S_t}b_{k,t}}_{\text{local drift}}, \text{where } b_{k,t} := \mathbb{E}_t[\bar{g}_{k,t}] - \nabla J_k(\theta_t).
\tag{9}
$$

We now bound the mean-squared norm of each contribution. (All bounds hold component-wise and hence for the Euclidean norm.)

**Lemma 1** (FPC: client-sampling variance). *Let $x_1, \ldots, x_K \in \mathbb{R}^d$, $\bar{x} = \frac{1}{K}\sum_k x_k$, and $S$ be a uniform size-$M$ sample without replacement with $|S| = M$ where $1 \leq M \leq K$. Then*

$$
\mathbb{E}\left\|\frac{1}{M}\sum_{k\in S}x_k - \bar{x}\right\|^2 = \frac{K-M}{M(K-1)}\cdot\frac{1}{K}\sum_{k=0}^{K-1}\|x_k - \bar{x}\|^2.
$$

*Proof.* Let $\Sigma := \frac{1}{K}\sum_{k=0}^{K-1}(x_k - \bar{x})(x_k - \bar{x})^\top$. This is the standard finite-population correction (Cochran, 1977): $\text{Cov}\left(\frac{1}{M}\sum_{k\in S}x_k\right) = \frac{K-M}{M(K-1)}\Sigma$. Taking trace on both sides yields the claim since $\mathbb{E}\|Z - \mathbb{E}Z\|^2 = \text{tr}\,\text{Cov}(Z)$. $\square$

Applying Lemma 1 with $x_k = \nabla J_k(\theta_t)$ and using $\nabla J(\theta_t) = \frac{1}{K}\sum_{k=0}^{K-1}\nabla J_k(\theta_t)$, we obtain

$$
\mathbb{E}_t\left\|\frac{1}{M}\sum_{k\in S_t}\nabla J_k(\theta_t) - \nabla J(\theta_t)\right\|^2 \leq \frac{K-M}{M(K-1)}\zeta^2.
\tag{10}
$$

**Lemma 2** (Local stochastic noise). *With the standing bounded-second-moment assumption, for each client $k$ and round $t$, $\mathbb{E}_t\|\bar{g}_{k,t} - \mathbb{E}_t\bar{g}_{k,t}\|^2 \leq G^2 + \sigma^2$. Moreover, conditioned on $\theta_t$ and $S_t$, the per-client noises are independent across $k \in S_t$. Consequently,*

$$
\mathbb{E}_t\left\|\frac{1}{M}\sum_{k\in S_t}\left(\bar{g}_{k,t} - \mathbb{E}_t\bar{g}_{k,t}\right)\right\|^2 \leq \frac{G^2 + \sigma^2}{M}.
\tag{11}
$$

*Proof.* Since $\bar{g}_{k,t} = \frac{1}{\tau}\sum_{i=0}^{\tau-1}g_{k,t,i}$ and $\mathbb{E}\|g_{k,t,i}\|^2 \leq G^2 + \sigma^2$, we have $\mathbb{E}_t\|\bar{g}_{k,t}\|^2 \leq G^2 + \sigma^2$, hence $\mathbb{E}_t\|\bar{g}_{k,t} - \mathbb{E}_t\bar{g}_{k,t}\|^2 \leq \mathbb{E}_t\|\bar{g}_{k,t}\|^2 \leq G^2 + \sigma^2$. Independence across clients (conditional on $\theta_t, S_t$) implies that variances add, yielding equation 11. $\square$

**Lemma 3** (Local drift/bias bound). *Let*

$$
b_{k,t} := \mathbb{E}_t[\bar{g}_{k,t}] - \nabla J_k(\theta_t), \qquad \bar{g}_{k,t} = \frac{1}{\tau}\sum_{i=0}^{\tau-1}g_{k,t,i},
$$

*where the local iterates satisfy $\theta_{k,t,0} = \theta_t$, $\theta_{k,t,i+1} = \theta_{k,t,i} + \eta\, g_{k,t,i}$, $\mathbb{E}[g_{k,t,i} \mid \theta_{k,t,i}] = \nabla J_k(\theta_{k,t,i})$, and $\mathbb{E}\|g_{k,t,i}\|^2 \le G^2 + \sigma^2$. If $J_k$ is L-smooth, then*

$$\mathbb{E}_t\|b_{k,t}\|^2 \;\le\; \frac{L^2\eta^2(\tau-1)^2}{4}\,(G^2 + \sigma^2). \tag{12}$$

*Moreover, for any sampled set $S_t$ of size $M$,*

$$\mathbb{E}_t\Big\|\frac{1}{M}\sum_{k\in S_t} b_{k,t}\Big\|^2 \;\le\; \frac{1}{M}\sum_{k\in S_t}\mathbb{E}_t\|b_{k,t}\|^2 \;\le\; \frac{L^2\eta^2(\tau-1)^2}{4}\,(G^2 + \sigma^2). \tag{13}$$

*Proof.* By definition and $L$-smoothness of $J_k$,

$$b_{k,t} \;=\; \frac{1}{\tau}\sum_{i=0}^{\tau-1}\big(\nabla J_k(\theta_{k,t,i}) - \nabla J_k(\theta_t)\big) \;=\; \frac{1}{\tau}\sum_{i=1}^{\tau-1} H_{k,t,i}\,(\theta_{k,t,i} - \theta_t),$$

where each $H_{k,t,i}$ is a (mean-value) linear map with operator norm $\|H_{k,t,i}\| \le L$. Using the local recursion $\theta_{k,t,i} - \theta_t = \eta\sum_{j=0}^{i-1} g_{k,t,j}$ and swapping sums gives

$$b_{k,t} \;=\; \frac{\eta}{\tau}\sum_{j=0}^{\tau-2}\Big(\sum_{i=j+1}^{\tau-1} H_{k,t,i}\Big) g_{k,t,j} \;=:\; \frac{\eta}{\tau}\sum_{j=0}^{\tau-2} A_{k,t,j}\, g_{k,t,j},$$

with $A_{k,t,j} := \sum_{i=j+1}^{\tau-1} H_{k,t,i}$ and hence $\|A_{k,t,j}\| \le \sum_{i=j+1}^{\tau-1}\|H_{k,t,i}\| \le L(\tau - 1 - j)$.

Applying the weighted Cauchy-Schwarz inequality,

$$\Big\|\sum_j A_{k,t,j} g_{k,t,j}\Big\|^2 \le \Big(\sum_j \|A_{k,t,j}\|\Big)\Big(\sum_j \frac{\|A_{k,t,j} g_{k,t,j}\|^2}{\|A_{k,t,j}\|}\Big) \le \Big(\sum_j \|A_{k,t,j}\|\Big)\Big(\sum_j \|A_{k,t,j}\|\,\|g_{k,t,j}\|^2\Big),$$

and taking $\mathbb{E}_t$ together with $\mathbb{E}\|g_{k,t,j}\|^2 \le G^2 + \sigma^2$ yields

$$\mathbb{E}_t\|b_{k,t}\|^2 \;\le\; \frac{\eta^2}{\tau^2}\Big(\sum_{j=0}^{\tau-2}\|A_{k,t,j}\|\Big)^2 (G^2 + \sigma^2) \;\le\; \frac{\eta^2}{\tau^2}\Big(L\sum_{j=0}^{\tau-2}(\tau - 1 - j)\Big)^2 (G^2 + \sigma^2).$$

Since $\sum_{j=0}^{\tau-2}(\tau - 1 - j) = \sum_{m=1}^{\tau-1} m = \frac{\tau(\tau-1)}{2}$, we obtain

$$\mathbb{E}_t\|b_{k,t}\|^2 \;\le\; L^2\eta^2\,\frac{(\tau-1)^2}{4}\,(G^2 + \sigma^2),$$

which is equation 12. For the client average, convexity of the squared norm (or Jensen) gives

$$\mathbb{E}_t\Big\|\frac{1}{M}\sum_{k\in S_t} b_{k,t}\Big\|^2 \;\le\; \frac{1}{M}\sum_{k\in S_t}\mathbb{E}_t\|b_{k,t}\|^2,$$

and the second inequality in equation 13 follows by applying equation 12 to each $k \in S_t$. □

With Lemma 3 in place, combining equation 10, equation 11, equation 13, equation 9, and Young's inequality $2\langle X, Y\rangle \le \|X\|^2 + \|Y\|^2$ gives the (assumption-free) second-moment control

$$\mathbb{E}_t\|e_t\|^2 \;\le\; \frac{G^2 + \sigma^2}{M} \;+\; \frac{2(K - M)}{M(K - 1)}\zeta^2 \;+\; \frac{L^2\eta^2(\tau-1)^2}{2}\,(G^2 + \sigma^2). \tag{14}$$

**C. Closing the one-step inequality.** Insert equation 14 into equation 8 and use $\eta = \frac{1}{L\tau}$ to get

$$\mathbb{E}_t[J(\theta_{t+1})] \;\ge\; J(\theta_t) + \frac{1}{2L}\|\nabla J(\theta_t)\|^2 - \frac{1}{2L}\Big[\frac{G^2 + \sigma^2}{M} + \frac{2(K - M)}{M(K - 1)}\zeta^2 + \frac{(\tau-1)^2}{2\tau^2}\,(G^2 + \sigma^2)\Big] \tag{15}$$

**D. PL inequality, recursion, and averaging.** Let $\delta_t := \mathbb{E}\big[J(\theta^\star) - J(\theta_t)\big]$. By the PL condition, $\|\nabla J(\theta_t)\|^2 \geq 2\mu\,\delta_t$. Taking total expectation of equation 15 and using $\eta = \frac{1}{L\tau}$ together with the variance-bias bound that includes the mixed-term control (i.e., $2\langle S, D\rangle \leq \|S\|^2 + \|D\|^2$), we obtain

$$\delta_{t+1} \leq \left(1 - \frac{\mu}{L}\right)\delta_t + \frac{1}{2L}\left[\frac{G^2 + \sigma^2}{M} + \frac{2(K-M)}{M(K-1)}\zeta^2 + \frac{(\tau-1)^2}{2\tau^2}(G^2 + \sigma^2)\right]. \quad (16)$$

Summing equation 16 over $t = 0, \ldots, T-1$ and dividing by $T$, and noting that $\sum_{t=0}^{T-1}(\delta_{t+1} - \delta_t) = \delta_T - \delta_0 \leq \delta_0$, yields

$$\frac{1}{T}\sum_{t=0}^{T-1}\delta_t \leq \frac{L}{\mu T}\delta_0 + \frac{1}{2\mu}\left[\frac{G^2 + \sigma^2}{M} + \frac{2(K-M)}{M(K-1)}\zeta^2 + \frac{(\tau-1)^2}{2\tau^2}(G^2 + \sigma^2)\right].$$

Finally, let $\tilde{\theta}$ be drawn uniformly from $\{\theta_t\}_{t=0}^{T-1}$. Then $\mathbb{E}[J(\theta^\star) - J(\tilde{\theta})] = \frac{1}{T}\sum_{t=0}^{T-1}\delta_t$, which gives the claimed bound. $\qquad\square$

