# OpenReview forum: "Federated Agent Reinforcement Learning"
_ICLR.cc/2026/Conference — Submitted to ICLR 2026_

### Official Review · Reviewer_LsvX · 2025-10-27

**Soundness:** 2
**Presentation:** 3
**Contribution:** 2
**Rating:** 2
**Confidence:** 4

**Summary:**

The paper proposes FedAgent, a federated reinforcement learning approach for training LLM-based agents without centralizing user data. The authors build FedAgentGym as a benchmark with 4 LLMs, 2 environments (WebShop, ALFWorld), and introduce three types of heterogeneity specific to agent learning. They provide theoretical convergence analysis and experiments showing FedAgent matches centralized performance.

The problem is genuinely important - training agents on sensitive user data is a real concern. The experimental setup is systematic and the heterogeneity characterizations are thoughtful. But I have serious concerns about the novelty claims, theoretical assumptions that aren't validated, and some experimental choices that make me question the conclusions.

**Strengths:**

1. **The problem motivation is solid.** Privacy in agent training is genuinely important and underexplored. With regulations like GDPR, we need solutions here.

2. **The heterogeneity partitioning strategies are clever.** PreferencePartition, CoveragePartition, and HardnessPartition are well-designed with single hyperparameters to control heterogeneity intensity while keeping other factors constant. This is good experimental design.

3. **Comprehensive evaluation across multiple dimensions.** Testing 4 LLMs, 2 environments, varying decentralized settings (samples per client, clients per round, epochs per round), and three heterogeneity types shows thoroughness.

4. **Strong robustness to heterogeneity.** The results in Figure 4 surprised me - FedAgent maintains stable performance even under extreme heterogeneity . This is actually pretty impressive and suggests the approach has practical promise.

**Weaknesses:**

### Critical

**1. The FedRLHF elephant in the room**

I keep coming back to this because it's just so glaring. Fan et al. 2025 publishes "FedRLHF: Federated Framework for Privacy-Preserving and Personalized RLHF" at AAMAS 2025. You cite it. But you never discuss it in the main text. How is FedAgent different? Here's what I need to see:

- Side-by-side comparison table of assumptions, algorithms, theoretical results
- Empirical comparison on at least one shared benchmark
- Clear positioning statement: "FedRLHF focuses on X, while FedAgent addresses Y"
- Discussion of why agent learning requires different treatment than general RLHF

Without this, the entire novelty claim falls apart. For all I know, you're just running FedRLHF's algorithm on agent tasks and calling it new.

**2. Unvalidated theoretical assumptions**

The PL condition (Assumption 4) needs empirical validation. This is straightforward: for a sample of your training runs, plot $\|\nabla J(\theta_t)\|^2$ versus $J(\theta^*) - J(\theta_t)$ on log-log scale. Do you see the required linear relationship? What's the estimated μ value?

Also, Assumption 5 (bounded heterogeneity) claims $\frac{1}{K}\sum_k \|\nabla J_k(\theta) - \nabla J(\theta)\|^2 \leq \zeta^2$. Can you compute actual $\zeta^2$ values for your different heterogeneity settings? This would connect the theory to your empirical heterogeneity measures.

Right now the theory section feels like it was included to check a box rather than actually inform the experimental work.

**3. Weak baseline and inconsistent error reporting**

The local training baseline is just wrong. You test clients 21, 42, 84 - why? How were these chosen? Are they representative of average, best, worst performance? Looking at Table 1, client 84 often does worse than others - is that typical or did you cherry-pick a weak client to make FedAgent look better?

A proper baseline would be:
- Average local training performance across all 100 clients (with std dev)
- Best and worst single-client performance
- Or stratified sampling across heterogeneity levels

Also: centralized and FedAgent report mean±std over 3 seeds, but local training shows single runs with no error bars. Why?.

### Significant Issues

**4. Missing key related work**

Your related work section is too sparse. You should discuss:
- Recent federated instruction tuning for LLMs (2023-2024 literature)
- Privacy-preserving LLM training methods beyond basic FL
- Distributed RLHF approaches
- Recent agent learning work that may touch on decentralized training

The comparison with traditional federated RL (Liu 2024, Qi 2021) is superficial. The seminar convergence analysis work for federated RL is missing from discussion, e.g, FedPG (Fan et al. NeurIPS 2021). You claim LLM agent learning has "fundamentally new challenges" but don't deeply analyze what's actually different. Multi-step reasoning and complex interactions exist in traditional RL too. Also as noted in issues#1, how is your paper fundamentally different from FedRLHF (Fan et al. AAMAS 2025)? Necessary discussed is needed.

**5. Privacy claims without privacy analysis**

Your entire motivation is privacy, yet you provide:
- No differential privacy guarantees
- No privacy-utility tradeoff analysis
- No discussion of gradient inversion attacks
- No membership inference analysis
- No secure aggregation protocols

There's extensive literature showing that model parameters leak training data information. You need at least a basic DP analysis or discussion of why DP isn't needed here.

**6. Severely limited practical grounding**

Some practical questions that aren't addressed:

- **Communication costs:** A single round of FedAgent with Qwen-7B requires each client to download ~14GB, train, and upload ~14GB. Over 70 rounds that's nearly 2TB per client. Is this realistic?

- **Who are the clients?** Individual users on smartphones? Organizations with their own data? This drastically changes the problem.

- **Incentive mechanisms:** Why would clients participate? Training LLMs locally is expensive.

- **Stragglers and dropouts:** What happens when clients disconnect? Your algorithm assumes all selected clients complete training.

- **Only evaluated on simulated benchmarks,** not real federated deployments. WebShop and ALFWorld are fine for initial evaluation but tell us nothing about real-world viability.

**7. Theory-practice gap**

Beyond the PL condition issue:

- Theorem 1 claims O(1/T) convergence. Can you verify this empirically? Plot the suboptimality gap versus rounds on log-log scale.

- The stepsize η=1/(Lτ) appears in the theory. What stepsize did you actually use in experiments? Was it tuned or derived from estimated L?

- The "noise floor" terms in Theorem 1 predict how variance depends on M, K, τ. Do your experiments validate these dependencies?

There's basically zero connection between your theory and experiments beyond "we have a theorem."

**8. Experimental design details**

Several important details are missing or unclear:

- **Client overlap:** You mention "potential overlap" but never quantify it. On average, how many clients share each instruction? This is absolutely critical for interpreting results.

- **Hyperparameter selection:** How were learning rates, batch sizes, etc. chosen? If you did cross-validation across clients, that leaks information and violates privacy. If you used a held-out test set, how was it created without central data access?

- **Why GRPO?** You use GRPO for policy optimization but don't justify this choice. Would PPO or A3C work similarly? Is GRPO particularly suited to federated settings?

- **Why FedAvg?** Simple model averaging is the most basic aggregation. FedProx handles heterogeneous objectives better. SCAFFOLD reduces client drift. Did you experiment with alternatives?

### Minor Issues (But Still Worth Fixing)

**9. Statistical rigor**

- No statistical significance testing. Are differences between FedAgent and centralized actually significant?
- Some error bars are quite large (e.g., ALFWorld Pick2 results) but there's no discussion of stability or variance.

**10. Claims that need verification**

- "First decentralized agent learning environment" - have you done a thorough search? This is a strong claim.

- "Fundamentally new challenges" - Your heterogeneity types are agent-specific but conceptually similar to existing FL heterogeneity (preference to label skew, coverage to quantity imbalance, hardness to quality differences). "Fundamentally new" is overstated unless discussed in depth.

11. Figure 2's circle marks for individual clients are hard to visually interpret.

**Questions:**

These are things where I wish to see in the manuscript:

1. **What's the relationship with FedRLHF?** Please provide a detailed comparison. If they're similar, why is a separate paper needed? If different, what are the key algorithmic or empirical differences?

2. **Can you validate the PL condition?** Plot gradient norm squared versus suboptimality gap to check if the linear relationship holds.

3. **Why clients 21, 42, 84 for local training?** Can you report aggregated statistics across all clients instead?

4. **What's the actual client overlap?** On average, how many clients share each instruction?

5. **How were hyperparameters selected without violating privacy?**

6. **What are the communication costs in GB?** Is this practical for real deployments?


7. **Figure 2b (ALFWorld) is way more volatile than WebShop. Why?** Does this affect your convergence guarantees?

---

> ### Author Response · Authors · 2025-12-03
> **Response to Reviewer LsvX (1)**
>
> We sincerely thank Reviewer LsvX for the detailed comments.
>
> ### First of all, we would like to clarify and emphasize the scope and core contributions of this work:
>  - **Our paper `made the first attempt to formally formulate and systematically analyze a new general decentralized agent learning paradigm`: foundation model based federated agent reinforcement learning (FedAgent).**
> - **We construct the `first decentralized agent learning environment` to analyze the performance of FedAgent systematically and controllably (FedAgentGym)**
> - **We propose `newly defined agent-specific heterogeneity challenges (Preference, Coverage, and Hardness Heterogeneity)`, and provides a `tailored theoretical convergence analysis`.**
> - **For the first time, we `empirically demonstrated the effectiveness` of federated foundation agent reinforcement learning `and its robustness` against the newly defined agent-specific heterogeneity challenges. We also provide insights on `its sensitivity to different decentralized settings.`**
> - **`Our work has opened a new direction for both fields of foundation agent learning and federated learning.` We release our code and environment as an extendable open-source library to inspire more future works in this new direction: https://anonymous.4open.science/r/federated_agent_submission-4652**
>
> > Comparison with FedRLHF
>
> We would like to point out that FedAgent and FedRLHF address completely different problems, heterogeneity dimensions, and environments.
>
> 1. **Problem formulation difference.**
>    FedRLHF addresses **federated preference-based fine-tuning** (reward model training + policy refinement) for general LLMs.
>    **Our work addresses federated *multi-step agent reinforcement learning* with environment interaction**, which is fundamentally different both algorithmically and in data generation.
>    * Our setting incorporates **environment transitions, multi-step trajectories, tool-use, and natural language reasoning**.
>    * RLHF has no environment transitions or multi-step rollouts.
>
> 2. **New heterogeneity dimensions specific to agent RL.**
>    We introduce **Preference, Coverage, Hardness heterogeneity**, which do not appear in RLHF.
>
> 3. **Environment.**
>    FedAgentGym (four LLMs, WebShop + ALFWorld, three forms of heterogeneity, multiple decentralized factors) is the **first environment for decentralized agent RL**, which is not applicable for RLHF frameworks.
>
> > theoretical assumptions (PL condition, bounded heterogeneity)
>
> Line 1211-1215 clearly explain the rationales of adopting these assumptions (`In practice, policy-gradient methods that constrain update size, such as trust-region approaches or proximal policy methods, yield smoother policy updates, making the PL assumption
> more tenable. Recent works have likewise employed PL-type conditions to obtain convergence
> guarantees for non-convex reinforcement learning objective`)
>
> As mentioned in Line 1215-1218, previous literature also support the adoption of these assumptions. (`PL condition is widely adopted in previous works such as FedRLHF and (Bhandari & Russo, 2024; Karimi et al., 2016; Yuan et al., 2022). Bounded heterogeneity is widely adopted in the literature such as (Li et al., 2020; Karimireddy et al., 2020; Stich, 2018; Khaled et al., 2020; Woodworth et al., 2020).`)
>
> > baseline
>
> We follow the literature [1,2] to design "Centralized Agent Training and Local Agent Training" as the baselines. **Since this is the initial exploration of federated foundation agent reinforcement learning, we aim to explore whether the performance can be comparable to Centralized Agent Training and outperform Local Agent Training.**
>
> The client indexes for local training are randomly selected. Aggregation over all the 100 clients each round is not practical and costs too long time each round. The code and results are released in https://anonymous.4open.science/r/federated_agent_submission-4652 for verification.
>
>
> [1] A Survey on Federated Fine-tuning of Large Language Models https://arxiv.org/abs/2503.12016
>
> [2] Federated Large Language Models: Current Progress and Future Directions https://arxiv.org/abs/2409.15723

---

> ### Author Response · Authors · 2025-12-03
> **Response to Reviewer LsvX (2)**
>
> > related work
>
> We would like to point out that we have extensively compare with conventional federated learning paradigms and foundation agent training works.
>
> As discussed in Lin 39-47, we have discussed the differences from  conventional federated learning paradigms: `Supervised federated learning is usually built on static data distributions and one-shot predictions, while traditional FRL typically assumes simple rewards, low-dimensional state and action spaces. In contrast, LLM agent learning involves  natural language state and action spaces, diverse task formulations, and complex environment  interactions, which create entirely new challenges for federated paradigms.`
>
> As discussed in Lin 39-47, we have discussed the differences from previous foundation agent training works: `In previous works, real-world task queries and trajectories have been essential for training AI agents in practical applications. However, they are becoming increasingly difficult to acquire due to privacy concerns. Our work makes an initial effort to explore training AI agents without compromising user data privacy.`
>
> > Privacy claims without privacy analysis, limited practical grounding, Theory-practice gap.
>
> These points are beyond the scope of this paper. We follow the literature on federated learning of foundation models [1,2].
>
>
> > Experimental design details
>
> The details of client partition is described in Appendix B.1. Hyperparameters are described clearly in Experiment Setup of Section 6. Since this is the initial exploration of federated foundation agent reinforcement learning, we use GRPO and FedAvg because of their wide adoption. Other algorithms can be incorporated into our framework easily in the future.
>
> ### **`We would like to clarify that we did not focus on one specific federated reinforcement learning algorithm (e.g., a model aggregation, or policy optimization, or client clustering method). We made the first attempt to explore a general federated learning paradigm: foundation model based federated agent reinforcement learning (FedAgent).`** Any specific conventional federated reinforcement learning algorithm can be incorporated into this framework.
>
> > Statistical rigor
>
> We have reported mean±std over 3 seeds in Table 1. Some error bars are quite large due to the intrinsic nature of agent RL training.
>
> > Claims that need verification
>
> To the best of our knowledge before submission, our paper is the first work to explore federated foundation agent reinforcement learning.
>
> In Section 3.3, we have clearly discussed the distinct difference between our proposed newly defined agent-specific heterogeneity challenges (Preference, Coverage, and Hardness Heterogeneity) and conventional heterogeneity challenges.
>
> [1] A Survey on Federated Fine-tuning of Large Language Models https://arxiv.org/abs/2503.12016
>
> [2] Federated Large Language Models: Current Progress and Future Directions https://arxiv.org/abs/2409.15723

---

### Official Review · Reviewer_wc3o · 2025-10-30

**Soundness:** 3
**Presentation:** 3
**Contribution:** 2
**Rating:** 4
**Confidence:** 3

**Summary:**

This paper explores agent reinforcement learning in federated learning settings and constructs a comprehensive benchmark environment.

**Strengths:**

1. Generally, this paper is easy to follow.

2. The three types of heterogeneity (Preference, Coverage, Hardness) are well-motivated.

**Weaknesses:**

1. FedAgent essentially applies standard FedAvg with GRPO. Although the application domain is somewhat novel, the algorithm contribution is limited.

2. During the experiment, the authors only compare the proposed method with centralized and local training, ignoring comparisons with other federated reinforcement learning methods.

3. The paper proposes three types of heterogeneity (Preference, Coverage, Hardness), but this taxonomy is insufficient to cover the diverse heterogeneity challenges that exist in real-world federated agent learning.

**Questions:**

1. Have you experimented with more sophisticated federated optimization algorithms (FedProx, FedNova, SCAFFOLD)? Why is vanilla FedAvg sufficient for this setting?

2. The experiments use uniform random client selection. Have you explored other strategies (e.g., importance sampling, clustered selection) that might leverage the heterogeneity structure?

---

> ### Author Response · Authors · 2025-12-03
> **Response to Reviewer wc3o**
>
> We sincerely thank Reviewer wc3o for the detailed comments.
>
> ### First of all, we would like to clarify and emphasize the scope and core contributions of this work:
>  - **Our paper `made the first attempt to formally formulate and systematically analyze a new general decentralized agent learning paradigm`: foundation model based federated agent reinforcement learning (FedAgent).**
> - **We construct the `first decentralized agent learning environment` to analyze the performance of FedAgent systematically and controllably (FedAgentGym)**
> - **We propose `newly defined agent-specific heterogeneity challenges (Preference, Coverage, and Hardness Heterogeneity)`, and provides a `tailored theoretical convergence analysis`.**
> - **For the first time, we `empirically demonstrated the effectiveness` of federated foundation agent reinforcement learning `and its robustness` against the newly defined agent-specific heterogeneity challenges. We also provide insights on `its sensitivity to different decentralized settings.`**
> - **`Our work has opened a new direction for both fields of foundation agent learning and federated learning.` We release our code and environment as an extendable open-source library to inspire more future works in this new direction: https://anonymous.4open.science/r/federated_agent_submission-4652**
>
> > the algorithm contribution is limited.
>
> ### As mentioned above, we would like to clarify that our key contribution is not a specific federated learning algorithm (e.g., a model aggregation, or policy optimization, or client clustering method). **Our paper `made the first attempt to formally formulate and systematically analyze a new general decentralized agent learning paradigm`: foundation model based federated agent reinforcement learning (FedAgent).** Any specific federated learning algorithm can be incorporated into our framework.
>
> ### Then, we construct the **`first decentralized agent learning environment`** to analyze the performance of FedAgent systematically and controllably (FedAgentGym).
>
> ### Moreover, our other contributions are described above.
>
> > comparisons with other federated reinforcement learning methods
>
> We follow the literature [1,2] to design "Centralized Agent Training and Local Agent Training" as the baselines. **Since this is the initial exploration of federated foundation agent reinforcement learning, we aim to explore whether the performance can be comparable to Centralized Agent Training and outperform Local Agent Training.**
>
> ### **`We would like to clarify that we did not focus on one specific federated reinforcement learning algorithm (e.g., a model aggregation, or policy optimization, or client clustering method). We made the first attempt to explore a general federated learning paradigm: foundation model based federated agent reinforcement learning (FedAgent).`** Any specific conventional federated reinforcement learning algorithm can be incorporated into this framework.
>
> [1] A Survey on Federated Fine-tuning of Large Language Models https://arxiv.org/abs/2503.12016
>
> [2] Federated Large Language Models: Current Progress and Future Directions https://arxiv.org/abs/2409.15723
>
> > The paper proposes three types of heterogeneity (Preference, Coverage, Hardness), but this taxonomy is insufficient to cover the diverse heterogeneity challenges that exist in real-world federated agent learning.
>
> We made the initial effort to explore the robustness of FedAgent against three newly defined agent-specific heterogeneity challenges. Hope this could inspire more future works.
>
> > Have you experimented with more sophisticated federated optimization algorithms (FedProx, FedNova, SCAFFOLD)? Why is vanilla FedAvg sufficient for this setting?
> >
> > The experiments use uniform random client selection. Have you explored other strategies (e.g., importance sampling, clustered selection) that might leverage the heterogeneity structure?
>
> Since this is the initial exploration of federated foundation agent reinforcement learning, we use FedAvg and uniform random client selection because of their wide adoption. Other algorithms can be incorporated into our framework easily in the future.

---

### Official Review · Reviewer_ByMC · 2025-11-01

**Soundness:** 3
**Presentation:** 3
**Contribution:** 2
**Rating:** 4
**Confidence:** 3

**Summary:**

This paper proposes FEDAGENT, a decentralized federated agent reinforcement learning paradigm for collaboratively training AI agents across distributed clients without sharing local data. It introduces the first decentralized benchmark environment, FEDAGENTGYM, featuring four types of LLM agents (Qwen2.5-{1.5B, 3B, 7B}, Llama-3.2-3B), two application scenarios (WebShop, ALFWorld), and three decentralized control dimensions (samples per client, clients per round, local epochs per client). The authors define three new heterogeneity types unique to decentralized LLM-agent learning: (i) Preference Heterogeneity: clients prefer different task types; (ii) Coverage Heterogeneity: clients have different task sampling scopes; and (iii) Hardness Heterogeneity: clients face tasks of varying difficulty. The paper also provides a convergence theorem showing an $\mathcal{O}(1/T)$ rate to the optimal solution up to a noise and heterogeneity floor. Empirical results show that FEDAGENT matches centralized performance, outperforms local training, and remains robust under high heterogeneity.

**Strengths:**

- The paper opens a new research direction in federated agent reinforcement learning, combining decentralized federated learning, reinforcement learning, and large language model (LLM) research in a way not previously explored. It directly tackles a critical real-world challenge, training efficient AI agents without centralizing sensitive user data, making it highly relevant.

- The work presents a well-structured experimental setup that spans diverse models, tasks, and heterogeneity settings. The authors analyze multiple dimensions, including sensitivity to decentralized parameters and robustness under different heterogeneity conditions. The results are consistent, well-documented, and convincingly support the paper’s claims.

- The theoretical analysis is thorough and mathematically sound. The proofs are detailed and clearly presented.

- The exposition is clear and well-organized, with informative figures that effectively illustrate the framework and results. Experimental details are thorough and transparent, ensuring reproducibility and aiding understanding.

**Weaknesses:**

- The paper currently compares only against centralized and local training baselines. To ensure a fair and comprehensive evaluation, the authors should consider including additional federated reinforcement learning methods such as FedRLHF [1], FedPG [2], and Federated Actor-Critic [3].

- The Polyak–Łojasiewicz ( assumption 4) is extremely strong. In fact, even in the simpler single-agent reinforcement learning, the agent's objective satisfies only a weaker non-uniform Łojasiewicz inequality [4].  In the heterogeneous federated reinforcement learning setting, such a type of inequality is provably not satisfied by the global objective [2].

- The analysis is not novel and closely resembles what has been done in [1]

- The paper does not provide any sample complexity in its theoretical analysis.

- Client heterogeneity is simulated using Gaussian or Beta noise, which limits interpretability and realism. It would be more meaningful to define heterogeneity in ways that directly reflect user-level variations (e.g., real shopping logs or user interaction histories)

[1] Fan et al, FedRLHF: A Convergence-Guaranteed Federated Framework for Privacy-Preserving and Personalized RLHF, AAMAS 2025

[2] Labbi et al, On Global Convergence Rates for Federated Policy Gradient under Heterogeneous Environment, ARXIV 2025

[3] Zhu et al,  Single-Loop Federated Actor-Critic across Heterogeneous Environments, AAAI 2025

[4]  Mei et al, On the global convergence rates of softmax policy gradient, ICML 2020

**Questions:**

- The paper currently compares only against centralized and local training baselines. Could the authors include additional federated reinforcement learning methods such as FedRLHF [1], FedPG [2], or Federated Actor-Critic [3] to provide a fairer and more comprehensive evaluation?

- The Polyak–Łojasiewicz (PL) condition used in Assumption 4 is quite strong. Do the authors have any empirical or theoretical evidence that the global objective in their setting satisfies this condition? Even in standard single-agent reinforcement learning, the objective typically satisfies only a non-uniform Łojasiewicz inequality [4]. How do the authors justify applying a stronger uniform PL assumption to the heterogeneous federated RL setting, where such inequalities are provably violated [2]?

- The theoretical analysis does not include a sample complexity bound. Can the authors provide such an analysis?

- Would it be possible to define heterogeneity in ways that more directly reflect real-world user behavior?

[1] Fan et al, FedRLHF: A Convergence-Guaranteed Federated Framework for Privacy-Preserving and Personalized RLHF, AAMAS 2025

[2] Labbi et al, On Global Convergence Rates for Federated Policy Gradient under Heterogeneous Environment, ARXIV 2025

[3] Zhu et al,  Single-Loop Federated Actor-Critic across Heterogeneous Environments , AAAI 2025

[4]  Mei et al, On the global convergence rates of softmax policy gradient, ICML 2020

---

> ### Author Response · Authors · 2025-12-03
> **Response to Reviewer ByMC (1)**
>
> We sincerely thank Reviewer ByMC for the detailed comments.
>
> ### First of all, we would like to clarify and emphasize the scope and core contributions of this work:
>  - **Our paper `made the first attempt to formally formulate and systematically analyze a new general decentralized agent learning paradigm`: foundation model based federated agent reinforcement learning (FedAgent).**
> - **We construct the `first decentralized agent learning environment` to analyze the performance of FedAgent systematically and controllably (FedAgentGym)**
> - **We propose `newly defined agent-specific heterogeneity challenges (Preference, Coverage, and Hardness Heterogeneity)`, and provides a `tailored theoretical convergence analysis`.**
> - **For the first time, we `empirically demonstrated the effectiveness` of federated foundation agent reinforcement learning `and its robustness` against the newly defined agent-specific heterogeneity challenges. We also provide insights on `its sensitivity to different decentralized settings.`**
> - **`Our work has opened a new direction for both fields of foundation agent learning and federated learning.` We release our code and environment as an extendable open-source library to inspire more future works in this new direction: https://anonymous.4open.science/r/federated_agent_submission-4652**
>
> > Comparison with FedRLHF
>
> We would like to point out that FedAgent and FedRLHF address completely different problems, heterogeneity dimensions, and environments.
>
> 1. **Problem formulation difference.**
>    FedRLHF addresses **federated preference-based fine-tuning** (reward model training + policy refinement) for general LLMs.
>    **Our work addresses federated *multi-step agent reinforcement learning* with environment interaction**, which is fundamentally different both algorithmically and in data generation.
>    * Our setting incorporates **environment transitions, multi-step trajectories, tool-use, and natural language reasoning**.
>    * RLHF has no environment transitions or multi-step rollouts.
>
> 2. **New heterogeneity dimensions specific to agent RL.**
>    We introduce **Preference, Coverage, Hardness heterogeneity**, which do not appear in RLHF.
>
> 3. **Environment.**
>    FedAgentGym (four LLMs, WebShop + ALFWorld, three forms of heterogeneity, multiple decentralized factors) is the **first environment for decentralized agent RL**, which is not applicable for RLHF frameworks.
>
> > theoretical assumptions (PL condition)
>
> Line 1211-1215 clearly explain the rationales of adopting these assumptions (`In practice, policy-gradient methods that constrain update size, such as trust-region approaches or proximal policy methods, yield smoother policy updates, making the PL assumption
> more tenable. Recent works have likewise employed PL-type conditions to obtain convergence
> guarantees for non-convex reinforcement learning objective`)
>
> As mentioned in Line 1215-1218, previous literature also support the adoption of these assumptions. (`PL condition is widely adopted in previous works such as FedRLHF and (Bhandari & Russo, 2024; Karimi et al., 2016; Yuan et al., 2022). Bounded heterogeneity is widely adopted in the literature such as (Li et al., 2020; Karimireddy et al., 2020; Stich, 2018; Khaled et al., 2020; Woodworth et al., 2020).`)
>
> > additional federated reinforcement learning methods such as FedRLHF [1], FedPG [2], and Federated Actor-Critic [3],
>
> We follow the literature [1,2] to design "Centralized Agent Training and Local Agent Training" as the baselines. **Since this is the initial exploration of federated foundation agent reinforcement learning, we aim to explore whether the performance can be comparable to Centralized Agent Training and outperform Local Agent Training.**
>
> ### **`We would like to clarify that we did not focus on one specific federated reinforcement learning algorithm (e.g., a model aggregation, or policy optimization, or client clustering method). We made the first attempt to explore a general foundation model based federated learning paradigm: federated agent reinforcement learning (FedAgent).`** Any specific conventional federated reinforcement learning algorithm can be incorporated into this framework.
>
> [1] A Survey on Federated Fine-tuning of Large Language Models https://arxiv.org/abs/2503.12016
>
> [2] Federated Large Language Models: Current Progress and Future Directions https://arxiv.org/abs/2409.15723
>
> > sample complexity in its theoretical analysis.
>
> Thanks for the suggestion. This point is **`explicitly beyond the scope of this initial exploration`**. We will explore it further in the future.

---

> ### Author Response · Authors · 2025-12-03
> **Response to Reviewer ByMC (2)**
>
> > Client heterogeneity is simulated using Gaussian or Beta noise, which limits interpretability and realism. It would be more meaningful to define heterogeneity in ways that directly reflect user-level variations (e.g., real shopping logs or user interaction histories)
>
> We would like to emphasize that our approach of client partition allows precise control over client distributions on preference, or coverage, or hardness heterogeneity with a hyperparameter, while maintaining the other impacting factors the same. **In this way, we can isolate and analyze the impact of each form of heterogeneity on FedAgent separately.**
>
> Thanks for the suggestion on defining heterogeneity in ways that directly reflect user-level variations. We will explore it further in the future.

---

### Official Review · Reviewer_iZBy · 2025-11-03

**Soundness:** 3
**Presentation:** 3
**Contribution:** 2
**Rating:** 4
**Confidence:** 4

**Summary:**

This paper introduces FedAgent, a framework for federated reinforcement learning (FRL) applied to LLM-based agents. The goal is to train multiple agents collaboratively without sharing raw data. The authors also build FedAgentGym as an experimental environment including several LLMs (Qwen and Llama series), two benchmarks (WebShop and ALFWorld), and configurable decentralized settings. They propose three new types of heterogeneity in terms of Preference, Coverage, and Hardness to study how client diversity affects learning. Experiments show that FedAgent achieves performance close to centralized training while maintaining data privacy.

**Strengths:**

+ Extends federated learning to LLM-agent reinforcement learning.
+ FedAgentGym provides a unified testbed for studying decentralized agent training with multiple models and environments.
+ Experiments systematically study decentralization parameters and heterogeneity factors.

**Weaknesses:**

- Algorithmic novelty is low, since the method is essentially FedAvg with policy-gradient updates.
- There is no regret or sample-complexity analysis.  The theoretical analysis directly follows standard federated SGD convergence proofs, but not efficiency or optimality results.
- the authors assume the global objective satisfies the PL condition. This allows them to show that the federated updates converge at rate O(1/T).  However, for LLM-based reinforcement learning, this assumption might not be realistic, since the rewards are highly non-smooth and multiple local optima might exist.
-  There is no study of system aspects, such as communication cost, latency, or client synchronization.  So the possible privacy and scalability advantages mentioned are not quantified.

**Questions:**

- Can you provide insight into communication overhead and how it scales with LLM size?  Are there compression or parameter-efficient alternatives?
- Can you provide empirical evidence supporting the PL or smoothness assumptions for LLM-based policies?
- Could the framework extend to multi-modal or embodied environments beyond WebShop and ALFWorld?  Or is this out of scope?
- Could you provide a regret or sample-complexity analysis to strengthen the theoretical foundation?

---

> ### Author Response · Authors · 2025-12-03
> **Response to Reviewer iZBy**
>
> We sincerely thank Reviewer iZBy for the detailed comments.
>
> ### First of all, we would like to clarify and emphasize the scope and core contributions of this work:
>  - **Our paper `made the first attempt to formally formulate and systematically analyze a new general decentralized agent learning paradigm`: foundation model based federated agent reinforcement learning (FedAgent).**
> - **We construct the `first decentralized agent learning environment` to analyze the performance of FedAgent systematically and controllably (FedAgentGym)**
> - **We propose `newly defined agent-specific heterogeneity challenges (Preference, Coverage, and Hardness Heterogeneity)`, and provides a `tailored theoretical convergence analysis`.**
> - **For the first time, we `empirically demonstrated the effectiveness` of federated foundation agent reinforcement learning `and its robustness` against the newly defined agent-specific heterogeneity challenges. We also provide insights on `its sensitivity to different decentralized settings.`**
> - **`Our work has opened a new direction for both fields of foundation agent learning and federated learning.` We release our code and environment as an extendable open-source library to inspire more future works in this new direction: https://anonymous.4open.science/r/federated_agent_submission-4652**
>
>
> > Algorithmic novelty is low
>
> ### As mentioned above, we would like to clarify that our key contribution is not a specific federated learning algorithm (e.g., a model aggregation, or policy optimization, or client clustering method). **Our paper `made the first attempt to formally formulate and systematically analyze a new general decentralized agent learning paradigm`: foundation model based federated agent reinforcement learning (FedAgent).** Any specific federated learning algorithm can be incorporated into our framework.
>
> ### Then, we construct the **`first decentralized agent learning environment`** to analyze the performance of FedAgent systematically and controllably (FedAgentGym).
>
> ### Moreover, our other contributions are described above.
>
> > theoretical assumptions (PL condition)
>
> Line 1211-1215 clearly explain the rationales of adopting these assumptions (`In practice, policy-gradient methods that constrain update size, such as trust-region approaches or proximal policy methods, yield smoother policy updates, making the PL assumption
> more tenable. Recent works have likewise employed PL-type conditions to obtain convergence
> guarantees for non-convex reinforcement learning objective`)
>
> As mentioned in Line 1215-1218, previous literature also support the adoption of these assumptions. (`PL condition is widely adopted in previous works such as FedRLHF and (Bhandari & Russo, 2024; Karimi et al., 2016; Yuan et al., 2022). Bounded heterogeneity is widely adopted in the literature such as (Li et al., 2020; Karimireddy et al., 2020; Stich, 2018; Khaled et al., 2020; Woodworth et al., 2020).`)
>
> > regret or sample-complexity analysis, system aspects, communication overhead, multi-modal or embodied environments beyond WebShop and ALFWorld
>
> Thanks for the suggestion. These points are **`explicitly beyond the scope of this initial exploration`**. We will explore these topics further in the future.

---

### Meta-Review · Area_Chair_xw5B · 2026-01-04

**Summary:**

This paper studied federated agent reinforcement learning, where a new algorithm is proposed with theoretical guarantees. It also provides empirical evaluation for the proposed algorithm.

The reviewers acknowledge that the problem setup is novel. However, they raise multiple concerns. In particular, a key limitation is the limited novelty of the theoretical analysis, which is built on strong assumptions such as the PL condition and bounded gradients, and largely follows existing work. Moreover, the analysis does not fully complete the proof: the second term in the convergence upper bound is non-vanishing, indicating that the algorithm does not converge to a stationary point. In addition, the analysis does not demonstrate how the number of agents and the communication period affect the convergence rate. These concerns were not adequately addressed in the rebuttal.

Due to these key limitations, this paper is recommended for rejection.

**Reviewer Concerns:**

They raise multiple concerns. In particular, a key limitation is the limited novelty of the theoretical analysis, which is built on strong assumptions such as the PL condition and bounded gradients, and largely follows existing work. Moreover, the analysis does not fully complete the proof: the second term in the convergence upper bound is non-vanishing, indicating that the algorithm does not converge to a stationary point. In addition, the analysis does not demonstrate how the number of agents and the communication period affect the convergence rate. These concerns were not adequately addressed in the rebuttal.

**Reviewer Scores:**

These concerns were not adequately addressed in the rebuttal. Therefore, the rating does not change.

---

### Decision · Program_Chairs · 2026-01-26

Reject